# MoDE: Weight Denoising Towards Better LLM Performance through a Mixture of Domain Experts

## Abstract

Pruning in large language models (LLMs) is widely assumed to degrade performance, since most weights are considered essential contributors to model capacity; thus, existing methods primarily rely on training to retain accuracy. However, our findings show that weight importance is domain-dependent rather than globally consistent, revealing the existence of noise weights whose removal can enhance domain-specific performance. To this end, we first present the DENoise (Domain Expert weight deNoising) algorithm, which effectively removes domain-aware noise weights without requiring fine-tuning to achieve improvement; We further develop MoDE (Mixture of Domain Experts), which treats these in-domain optimal denoised models as experts and employs a bilevel trainable router to dynamically activate them, thereby enhancing out-of-domain generalization. Experimental results show that applying the DENoise algorithm yields 2–3% gains across benchmarks such as MMLU, MBPP, and GSM8K, while MoDE achieves an average improvement of over 1.1% against baseline models, all without introducing additional parameters or tuning overhead.

## 1 Introduction

The rapid growth of large language models has made deployment increasingly costly due to excessive parameter counts (Patterson et al., 2021; Strubell et al., 2019). Pruning, which removes redundant weights while minimizing accuracy loss, is widely regarded as an effective solution(Cheng et al., 2024). A common strategy involves ranking parameters by importance—typically measured by magnitude (Han et al., 2015b; See et al., 2016), norm(Li et al., 2016; He et al., 2018), or estimated loss change(You et al., 2019; Liu et al., 2021)—and removing the lowest-ranked ones to preserve key contributing parameters to performance. However, these methods focus on identifying weights that are detrimental under global, general-purpose evaluation. As a result, when transferred to domain-specific scenarios, they may prune weights that are actually beneficial within the target domain, resulting in performance degradation.

However, this global perspective assumes domain invariance, whereas weight importance is in fact domain-specific. Our analysis shows that domain shifts can lead to gradient reversal, causing some weights that appear neutral under global training to exhibit negative contributions within specific tasks. Consequently, weights that occupy the middle ranges of global importance rankings may *act as domain-specific **Noise Weights**, and selectively removing them can in fact yield performance gains*. Informed by theoretical findings, we performed empirical pruning experiments and observed that removing weights from certain mid-to-low importance intervals led to improved accuracy, and was consistently observed across up to 57 tasks.

To achieve model enhancement, we integrate it into a Mixture-of-Experts (MoE) framework—whose core principle is that, for each specific domain of task, the model dynamically activates the most suitable expert. Following this rationale, we propose the DENoise (Domain Expert weight deNoising) algorithm, which identifies and removes noise-like weights (defined as Noise Weights) from self-attention and FFN layers to construct domain expert subnetworks. Building on this, we develop the MoDE (Mixture of Domain Expert) framework, which routes each input to its most relevant denoised

expert at the sequence level for efficient activation. Thereby, the main contributions of this work are as follows:

(i) We attempt to identify and the **Noise Weights** in LLM, which tentatively suggests that large language models may contain weights interfering with in-domain performance. This perspective complements existing importance-based pruning strategies and opens a speculative yet valuable theoretical dimension for understanding and approaching model compression.

(ii) We develop the **tuning-free DENoise algorithm**, which identifies and removes potential noise weights to derive task-specialized expert subnetworks from pretrained LLMs. Experimental results show an average performance improvement of 1%–3% on benchmarks such as MMLU, MBPP, and GSM8K, with a maximum gain of 6.8%.

(iii) We introduce **Mixture-of-Domain-Expert** (MoDE), which integrates the DENoise algorithm with a bilevel dynamic router to construct a domain expert system. MoDE requires no extensive model modifications and can be directly integrated into existing architectures, achieving an average performance improvement of 1.1% across multiple mainstream LLMs.

## 2 PRELIMINARIES

In this section, we introduce a perspective on weight evaluation in pretrained language models. We argue that weight importance is *domain-dependent*; that certain weights may exhibit *negative marginal contributions* on specific tasks, forming what we call *noise weights*; and that, under this view, traditional global pruning paradigms may face structural limitations in multi-domain settings.

### 2.1 OBSERVING WEIGHT IMPORTANCE IN PRETRAINED MODELS

**Claim.** Most pruning methods implicitly assume that weight importance can be captured by a global ordering, thereby inducing a universally shared sparse structure across tasks.

Let $w \in \mathbb{R}^n$ denote the model weights, and let $\mathcal{Q}$ denote a data distribution. The expected loss under $\mathcal{Q}$ is:

$$\mathcal{L}_{\mathcal{Q}}(w) = \mathbb{E}_{(x,y)\sim\mathcal{Q}}\big[\ell(f_w(x), y)\big], \tag{1}$$

where $f_w$ denotes the model parameterized by $w$ and $\ell$ is the per-example loss. For a given weight index $i$, define the *weight removal operation* as $w^{(-i)} = w - w_i e_i$, where $e_i$ is the $i$th basis vector. The loss change induced by removing weight $i$ is:

$$\Delta\mathcal{L}_{\mathcal{Q}}(i) = \mathcal{L}_{\mathcal{Q}}(w^{(-i)}) - \mathcal{L}_{\mathcal{Q}}(w). \tag{2}$$

Following standard pruning literature, the importance of weight $i$ under $\mathcal{Q}$ is defined as:

$$I_{\mathcal{Q}}(i) = -\Delta\mathcal{L}_{\mathcal{Q}}(i). \tag{3}$$

Early and recent pruning methods (e.g., (He et al., 2019; Zhao & Long, 2023; Frantar & Alistarh, 2023b)) approximate $I_{\mathcal{Q}}(i)$ using proxy scores such as weight magnitude or activation-based metrics. These approaches are used to construct a global importance ordering that is shared across all tasks, and consequently *much of the pruning literature focuses on determining how to prune while minimizing degradation to the model's overall general performance*.

### 2.2 WEIGHT CONTRIBUTIONS DIFFER ACROSS DOMAINS

**Claim.** Weight importance is not universal but varies across downstream domains as implied in the multi-domain nature of pretraining.

Let the pretraining corpus be composed of multiple sub-domains $\{P_k\}$, each associated with a mixture coefficient $\lambda_k$ satisfying $\lambda_k \geq 0$ and $\sum_k \lambda_k = 1$. The overall pretraining distribution is therefore defined as: $P_{\mathrm{mix}} = \sum_k \lambda_k P_k$. The model is trained to minimize the expected loss under this mixture: $\mathcal{L}_{P_{\mathrm{mix}}}(w) = \mathbb{E}_{(x,y)\sim P_{\mathrm{mix}}}\left[\ell(f_w(x), y)\right]$, where $f_w$ is the model parameterized by weights $w$ and $\ell$ is the per-example loss function. Then the importance of weight $i$ $I_{\mathrm{mix}}(i)$ under the pretraining mixture distribution becomes:

$$I_{\mathrm{mix}}(i) = -\Delta\mathcal{L}_{P_{\mathrm{mix}}}(i) = \mathcal{L}_{P_{\mathrm{mix}}}(w) - \mathcal{L}_{P_{\mathrm{mix}}}(w^{(-i)}), \tag{4}$$

where $w^{(-i)}$ denotes the weight vector with the $i$th weight removed. For a downstream domain $D$, the relevant contribution measure is:

$$I_D(i) = -\Delta\mathcal{L}_D(i) = \mathcal{L}_D(w) - \mathcal{L}_D(w^{(-i)}). \tag{5}$$

Then the *domain-induced importance shift* can be defined as:

$$\Delta I(i) = I_D(i) - I_{\text{mix}}(i). \tag{6}$$

A positive shift $\Delta I(i) > 0$ indicates that weight $i$ is more useful in the target domain $D$ than in the pretraining mixture, whereas a negative shift $\Delta I(i) < 0$ suggests reduced or even harmful contribution within that domain. It also indicates that the importance of a weight may shift across domains, and in particular, some weights may exhibit $I_D(i) < I_{\text{mix}}(i)$, or even $I_D(i) < 0$.

> **Insight 1.** *This suggests that pruning decisions need not always aim to avoid harm—under domain shifts, selectively removing certain weights can in fact lead to performance gains.*

### 2.3 NOISE WEIGHTS: NEGATIVE MARGINAL EFFECTS IN A DOMAIN

**Claim.** Weights that *reduce* task loss when removed naturally correspond to *noise weights*. Under domain shift, such weights arise from changes in local loss geometry, and their effect can be directly characterized using domain-specific weight importance.

Let the domain-specific loss be $\mathcal{L}_D(w) = \mathbb{E}_{(x,y)\sim D}\big[\ell(f_w(x), y)\big]$. Removing weight $w_i$ produces a loss change $\Delta\mathcal{L}_D(i) = \mathcal{L}_D(w - w_i e_i) - \mathcal{L}_D(w)$, where $e_i$ is the standard basis vector. The corresponding domain-specific importance is defined as

$$I_D(i) = \mathcal{L}_D(w) - \mathcal{L}_D(w - w_i e_i) = -\Delta\mathcal{L}_D(i) \approx g_i w_i - \frac{1}{2} h_{ii} w_i^2, \tag{7}$$

with applying the classical second-order Taylor expansion (LeCun et al., 1989; Hassibi et al., 1993), where $g_i = \frac{\partial \mathcal{L}_D(w)}{\partial w_i}$ and $h_{ii} = \frac{\partial^2 \mathcal{L}_D(w)}{\partial w_i^2}$. The condition for a weight to have *negative marginal contribution* (i.e., removing the weight *reduces* loss on $D$) becomes: $g_i w_i > \frac{1}{2} h_{ii} w_i^2$. Since $I_D(i) = -\Delta\mathcal{L}_D(i)$, a negative marginal contribution is equivalent to *positive importance*: removing weight $i$ reduces the loss on domain $D$ if and only if $I_D(i) > 0$. We therefore define the domain-specific noise weight set using importance:

$$\mathcal{N}_D = \{\, i \mid I_D(i) > 0 \,\}. \tag{8}$$

This importance-based formulation reveals that noise weights arise whenever domain-induced gradient reversal makes a previously helpful weight under $P_{\text{mix}}$ become detrimental within domain $D$. Detailed explanation see Appendix E.

### 2.4 MoE AS A MECHANISM FOR CROSS-DOMAIN GENERALIZATION

While $\mathcal{N}_D$ characterizes which weights are harmful within a specific domain, removing these weights globally may cause the model to lose useful capacity for other domains. This creates a structural tension: *domain-aware pruning improves the target domain but may compromise the mixture-level generality learned during pretraining*.

Thus, the central challenge is not only to detect domain-specific negative contributions, but to *allocate* weights in a way that preserves multi-domain competence. This motivates the use of a mixture-of-experts (MoE) formulation: instead of physically removing weights, we let different domains activate different effective subnetworks derived from the same pretrained parameters.

In this perspective, pruning results such as $\mathcal{N}_D$ become *routing signals* rather than deletion rules. They indicate which subsets of weights should be attenuated for domain $D$, while allowing other domains to retain or even rely on them. Consequently, domain-induced importance shifts can be leveraged to construct domain-specialized experts without sacrificing the model's global capabilities.

## 3 METHOD

### 3.1 STARTUP: A PROXY TO $\mathcal{N}_D$

Since exact domain-specific importance is computationally intractable at the scale of large language models, we resort to a scalable proxy and study how its induced ranking can be used to approximate the domain-specific noise set $\mathcal{N}_D$.

**Global Importance Proxy.** While domain importance provides a principled characterization of weight contribution, computing it exactly requires evaluating the loss change for each individual weight, which is infeasible for models with billions of parameters. Thus, practical pruning methods rely on proxy scores that preserve structural information about how strongly each weight participates in the forward computation. Among existing pruning literature, the magnitude–activation product

$$|W_i| \cdot \|X_i\|_2, \tag{9}$$

is widely used due to its computational efficiency and its ability to reflect how strongly weight $i$ is activated under the (mixture-level) pretraining distribution. In this sense, $|W_i| \cdot \|X_i\|_2$ provides a *global* importance ordering over weights.

**Domain-Aware Proxy to $\mathcal{N}_D$.** Although $|W_i| \cdot \|X_i\|_2$ is a global proxy, its correlation with the weight–activation terms in the Taylor expansion causes the extremes of the ranking to be universally stable while the mid-ranked region becomes most sensitive to domain-induced importance shifts, making it the most likely location of $\mathcal{N}_D$. Detailed explanation see Appendix E.

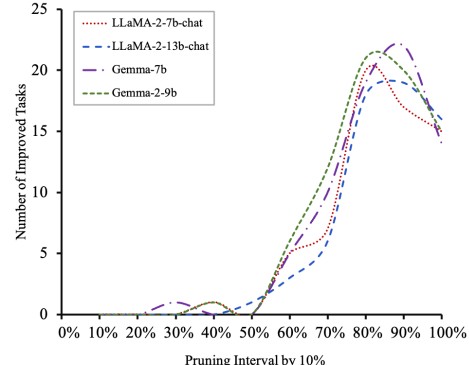

Figure 1: The number of tasks achieving accuracy improvements by 10% pruning across models.

To validate this connection, we examine pruning intervals along the proxy ranking on `LLaMA-2-7B/13B-chat` and `Gemma-7B/Gemma2-9B` across 57 tasks. Figure 1 shows that:

(1) many tasks improve in several mid-to-low importance regions;

(2) this pattern is consistent across all models;

(3) the beneficial intervals differ across tasks.

These behaviors match the condition $\Delta\mathcal{L}_D(i) < 0$, indicating that negative-contribution weights are widespread. Importantly, what aligns with domain importance is not the proxy values themselves but the *mid-ranked intervals* where pruning yields domain-specific gains—unlike conventional magnitude pruning, which removes only the lowest-scoring tail.

> **Insight 2.** *Our findings suggest that domain-specific negative-contribution weights do not necessarily lie at the extremes of a global ranking; instead, they often concentrate in mid-range intervals that become apparent only when evaluated in a domain-aware manner.*

### 3.2 DENOISE

Given the domain-dependent importance patterns established above, a practical method is needed to operationalize the identification and removal of domain-specific noise weights. This section introduces DENoise, which identifies domains, computes importance metrics, and derives a domain-specific expert through denoising.

**Domain-based Identification** Before performing noise-weight removal, we first determine the semantic domain (Hendrycks et al., 2021; Austin et al., 2021; Cobbe et al., 2021), to which a given task belongs, ensuring that subsequent domain-specific thresholding and importance estimation are computed on consistent statistical bases. To this end, we construct a set of coarse-grained knowledge domains $\mathcal{D} = \{D_1, D_2, \ldots, D_n\}$, where each $D_i$ corresponds to a dataset representing a distinct source of domain knowledge.

Given a task $\mathcal{T}$, we compute its posterior probability over main domains. Specifically, we extract the task embedding $F_\mathcal{T}$ using the last-layer hidden states of a pretrained Transformer, where the final embedding is obtained via mean-pooling over all token representations (Reimers & Gurevych, 2019). We compute the cosine similarity to each domain prototype vector $\mu_j$ using the standard cosine similarity formula, then similarities are normalized with a softmax: $P(D_j \mid \mathcal{T}) = \text{softmax}_j \left( \frac{F_\mathcal{T}^\top \mu_j}{\|F_\mathcal{T}\| \|\mu_j\|} \right)$. Thus, the main domain assignment is:

$$D_j = \arg\max_j P(D_j \mid \mathcal{T}). \tag{10}$$

To capture intra-domain variation, we further construct fine-grained subdomains. We collect all samples in $D_j$, extract their last-layer embeddings to form $\mathcal{F}_j$, and apply K-means clustering:

$$\mathcal{C}^* = \arg\min_\mathcal{C} \sum_{i=1}^{k} \sum_{f \in C_i} \|f - \mu_i\|_2. \tag{11}$$

To determine the optimal number of subdomains $k$, we compute the Silhouette Score for each candidate value (Rousseeuw, 1987; Petukhova et al., 2025). For a sample $i$, the Silhouette Score is defined as: $s(i) = \frac{b(i)-a(i)}{\max(a(i), b(i))}$, where $a(i)$ denotes the average distance between $i$ and all points within its assigned cluster, and $b(i)$ denotes the minimum average distance.

For a new task $\tilde{\mathcal{T}}$, we extract its last-layer embedding $F_{\tilde{\mathcal{T}}}$ and compute its distance to subdomain centroids:

$$\tilde{k} = \arg\min_k \|F_{\tilde{\mathcal{T}}} - \mu_k\|_2. \tag{12}$$

The task is thus assigned to the most semantically relevant subdomain $D_{j_{\tilde{k}}}$, which provides the appropriate local knowledge structure for downstream denoising.

$I_D$ **Identification.** After identifying the subdomain $D_{j_{\tilde{k}}}$, the next step is to quantify how each weight contributes within this subdomain so that domain-specific noise weights $\mathcal{N}_D$ can be detected.

First, to obtain a stable and scalable proxy for the domain-specific importance $I_D(i)$ without computing explicit loss differences, we aggregate local weight–activation interactions through patching. Since activation values indicate how strongly a weight responds to input features, We downsample the weight matrix $\mathcal{W}$ and activation matrix $X$ into a $\beta u \times \beta v$ grid ($\beta = 1/\alpha$), where each cell forms a local patch $P_{\eta\nu}$. For each patch, we compute a local importance score:

$$\mathcal{M}_{\eta\nu} = \sum_{(s,t) \in P_{\eta\nu}} \left( |\mathcal{W}_{st}| \cdot \|X_{st}\|_p \right), \tag{13}$$

where larger weights and stronger activations jointly indicate a higher contribution to the model output. This patch-level metric provides a more stable approximation of weight importance conditioned on the current subdomain.

$\mathcal{N}_D$ **Denoising.** After calculating patch importance scores, we determine candidate threshold intervals $[\theta_{\min}^{(i)}, \theta_{\max}^{(i)}]$ for each row based on the row-wise distribution of importance values. All weights whose patch importance falls within these intervals are treated as potential noise candidates:

$$\theta_r = \bigcup_{i=1}^{\beta u} \left\{ \mathcal{M}_{i:}(\sigma) \mid \mathcal{M}_{i:}(\sigma) \in [\theta_{\min}^{(i)}, \theta_{\max}^{(i)}] \right\}. \tag{14}$$

We then evaluate the effect of removing each candidate interval on the subdomain performance $\text{Acc}(\mathcal{W} \setminus \theta_r)$, and select the interval that yields the best performance after pruning:

$$\theta^* = \arg\max_{\theta_r \subseteq \mathcal{W}} \text{Acc}(\mathcal{W} \setminus \theta_r). \tag{15}$$

This search procedure acts as a practical approximation to the theoretical positive-importance condition $I_D(i) > 0$: if removing a set of weights improves performance in the subdomain, these weights exhibit negative marginal contribution and therefore belong to $\mathcal{N}_D$. Consequently,

$$\theta^* \approx \mathcal{N}_D, \qquad \mathcal{E}_\mathcal{T} = \mathcal{W} \setminus \theta^*, \tag{16}$$

where $\mathcal{E}_\mathcal{T}$ denotes the denoised parameter subset for the task $\mathcal{T}$. Finally, by aggregating all denoised hidden layers, we construct a task-specific *Domain Expert* that performs best under the domain-specific task.

Figure 2: The inference flow through our MODE. The *embedded input tokens* (⬜) is first processed by a bilevel trainable (🔥) router, where it is initially classified into the main knowledge domain by level 1 and further into a sub knowledge domain by level 2. The input is then passed to the identified Domain Expert, which is formed by applying the result from DENoise Algorithm, from the hidden layers of a frozen (❄) pre-trained LLM. Finally *output tokens* (🟦) are processed by Domain Expert to output.

### 3.3 MODE ARCHITECTURE

After obtaining domain-specialized experts constructed by DENoise, we introduce MODE(Mixture of Domain Experts), a framework that dynamically assigns any input task to the most relevant main domain and subdomain through a bilevel trainable router. This allows the model to fully leverage the denoised parameter subsets $\mathcal{E}_{D_{b,k}}$ and activate only the domain-relevant expert during inference.

**A Bilevel Trainable Router.** To perform hierarchical domain classification, we introduce a bilevel trainable router $\mathcal{R}$, splitting the decision into two stages. The router takes as input the task embedding $F_{\bar{\mathcal{T}}}$, obtained by mean-pooling the last-layer hidden states of the pretrained Transformer. Formally, the hierarchical routing is given by: $D_{b,k} = \mathcal{R}_{\text{sub},b}\big(\mathcal{R}_{\text{main}}(F_{\bar{\mathcal{T}}})\big)$. The router therefore consists of two stages:

(1) a *main-domain classifier* $\mathcal{R}_{\text{main}}$ that maps the task embedding into a main domain $D_b$;

(2) a *subdomain classifier* $\mathcal{R}_{\text{sub},b}$, conditioned on the selected main domain, that further assigns the task to a subdomain $D_{b,k}$.

We optimize the router using the standard cross-entropy objective. For each task embedding $F_{\bar{\mathcal{T}}}$ with its one-hot label $y$, the loss is computed as $\mathcal{L}_{\mathcal{R}} = -\sum_{q=1}^{n} y_q \log P(D_q \mid F_{\bar{\mathcal{T}}})$, with $P(D_q \mid F_{\bar{\mathcal{T}}})$ representing the predicted probability assigned to domain $D_q$.

**MODE Inference.** Given input tokens $X_{in}$, the inference procedure proceeds in four steps:

*Step 1: Encode the input.* The model embeds the input tokens and computes Transformer hidden states. Mean pooling over the final hidden layer yields the task representation: $F_{\bar{\mathcal{T}}}$.

*Step 2: Route the task.* The task representation is passed through the bilevel router to select the most relevant main domain and subdomain: $D_{b,k} = \mathcal{R}(F_{\bar{\mathcal{T}}})$.

*Step 3: Load the domain expert.* Since all domain experts are precomputed offline using DENoise, inference does not recompute denoising. Instead, the model directly loads the corresponding domain-specific expert:$W_{\text{expert}} = \mathcal{E}_{D_{b,k}}$.

*Step 4: Execute the expert and produce the output.* The selected expert processes the input representation $\hat{Y} = \mathcal{E}_{D_{b,k}}(X_{in})$, and a linear layer followed by softmax generates the final prediction: $Y = \text{Softmax}(W_o \hat{Y})$.

As illustrated in Figure 2, MODE dynamically selects the most appropriate domain expert for each input, ensuring that only domain-relevant parameters are activated during inference and enabling efficient, task-specialized computation.

## 4 EXPERIMENT

### 4.1 SETUP

**Datasets.** We conduct experiments across three domains: general knowledge, code, and mathematics. For general knowledge, we use MMLU (Hendrycks et al., 2021). For code, we adopt MBPP (Austin et al., 2021) and HumanEval (Chen et al., 2021), and for mathematics, we use GSM8K (Cobbe et al.,

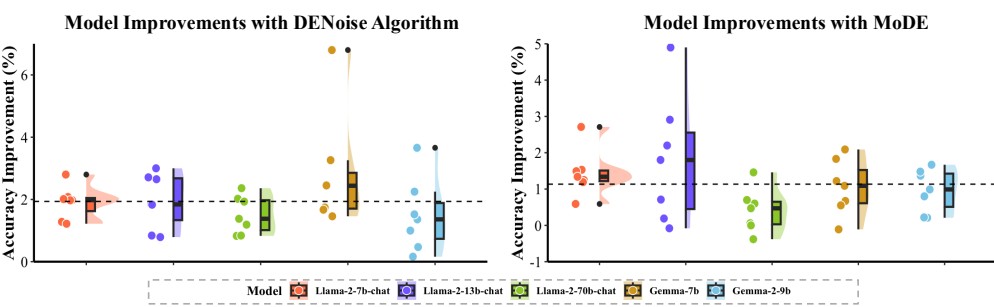

Figure 3: Comparative boxplot analysis of models using DENoise and MoDE methods.

2021) and MathQA (Amini et al., 2019). Validation sets from MMLU, MBPP, MathQA, and GSM8K are used as reference data, with final evaluations on their respective test sets. In addition, we include WinoGrande (Sakaguchi et al., 2021) and ARC-C (Clark et al., 2018) as out-of-distribution (OOD) evaluations to assess robustness beyond the primary domains. For HumanEval, we follow standard practice by using the MBPP validation set as reference and evaluating on the full dataset. We adopt a 5-shot for MMLU and 0-shot for all other datasets.

**Baseline & Foundation Models.** We evaluate our method on five foundation models: `llama-2-7b/13b/70b-chat` and `Gemma-7b/2-9b`, covering a wide range of model sizes. Both `LLaMA-2` and `Gemma` series are decoder-only transformer-based models optimized for dialogue and natural language understanding.

**Evaluation Metrics.** We report accuracy (acc%) for general and mathematics tasks. For coding tasks, we additionally use pass@1 and pass@10, which measure the probability that at least one of the top-$k$ generated solutions is correct, calculated as: pass@k $= \frac{1}{N}\sum_{i=1}^{N} c_i$, where $c_i \in \{0, 1\}$ indicates whether the $i$-th task has at least one correct solution in Top-$k$.

Table 1: Performance comparison of DENOISE and MODE with baseline on foundation models.

| | Method | MMLU | MBPP | | HumanEval | | GSM8K | MathQA | WinoGrande | ARC-C |
|---|---|---|---|---|---|---|---|---|---|---|
| | | acc | pass@1 | pass@10 | pass@1 | pass@10 | acc | acc | acc | acc |
| | BASELINE | 45.81 | 19.24 | 23.60 | 14.45 | 19.51 | 20.24 | 25.33 | 68.98 | 46.25 |
| | Std (5-run) | ±0.12 | ±0.15 | ±0.25 | ±0.12 | ±0.25 | ±0.15 | ±0.18 | ±0.24 | ±0.09 |
| llama-2-7b | DENOISE | 47.83 | 21.32 | 26.40 | 15.73 | 20.73 | 22.21 | 27.34 | − | − |
| | MODE | 47.34 | 20.58 | 25.80 | 14.51 | 19.51 | 20.79 | 26.13 | 70.21 | 47.93 |
| | BASELINE | 52.34 | 9.68 | 13.00 | 18.66 | 28.05 | 31.77 | 24.86 | 72.22 | 48.98 |
| | Std (5-run) | ±0.12 | ±0.16 | ±0.24 | ±0.11 | ±0.27 | ±0.16 | ±0.17 | ±0.21 | ±0.18 |
| llama-2-13b | DENOISE | 53.18 | 11.52 | 16.00 | 19.45 | 29.88 | 34.42 | 27.57 | − | − |
| | MODE | 52.93 | 12.39 | 14.80 | 19.13 | 29.27 | 33.86 | 26.34 | 75.01 | 50.65 |
| | BASELINE | 66.87 | 45.22 | 66.15 | 30.52 | 59.34 | 59.47 | 35.12 | 79.96 | 54.66 |
| | Std (5-run) | ±0.10 | ±0.14 | ±0.23 | ±0.12 | ±0.22 | ±0.13 | ±0.16 | ±0.11 | ±0.16 |
| llama-2-70b | DENOISE | 68.89 | 47.15 | 66.98 | 31.71 | 60.72 | 61.83 | 35.96 | − | − |
| | MODE | 68.12 | 45.93 | 66.34 | 31.22 | 60.01 | 60.56 | 35.33 | 81.53 | 57.43 |
| | BASELINE | 63.56 | 2.94 | 9.00 | 15.31 | 20.12 | 57.92 | 37.12 | 72.17 | 53.06 |
| | Std (5-run) | ±0.13 | ±0.18 | ±0.27 | ±0.14 | ±0.25 | ±0.17 | ±0.20 | ±0.17 | ±0.14 |
| Gemma-7b | DENOISE | 65.30 | 6.20 | 15.80 | 16.77 | 22.56 | 59.59 | 39.57 | − | − |
| | MODE | 65.05 | 5.85 | 13.90 | 12.93 | 18.29 | 59.29 | 38.79 | 74.41 | 54.17 |
| | BASELINE | 69.71 | 8.36 | 9.80 | 12.87 | 18.90 | 68.46 | 50.75 | 80.52 | 68.35 |
| | Std (5-run) | ±0.12 | ±0.15 | ±0.25 | ±0.13 | ±0.23 | ±0.16 | ±0.18 | ±0.08 | ±0.11 |
| Gemma-2-9b | DENOISE | 71.07 | 8.52 | 10.80 | 15.12 | 22.56 | 69.98 | 51.22 | − | − |
| | MODE | 70.90 | 8.28 | 10.40 | 14.33 | 20.73 | 69.45 | 50.97 | 81.78 | 69.98 |

## 4.2 MAIN RESULTS

The proposed DENOISE algorithm serves as a domain expert to each specific task, thereby achieving the best in-domain performance. When integrated into the MODE framework, DENOISE further enables MODE to exhibit robust out-of-domain generalization, ensuring adaptability, and stability across diverse benchmarks, as demonstrated in Table 1.

**DENOISE.** The application of DENOISE using $l_2$-norm and 10% de noise ratio results in consistent performance improvements with an average gain of 2.04% over the baseline models, as shown in Table 1. Specifically, `llama-2-7b-chat-hf`'s accuracy on MMLU increases by 2.02%, while its performance on MBPP (pass@1) rose by 2.08%, and its accuracy on GSM8K improves by 1.97%. Similar patterns are observed for the other models. Notably, `Gemma-7b` exhibits an increase of 3.26% in MBPP (pass@10) and a 2.4% improvement in MathQA accuracy, demonstrating DENOISE's efficacy across different models and tasks.

**MODE.** We evaluate MODE on the same set of LLMs and observe an average improvement of 1.11%, which—though smaller than DENOISE—remains stable and effective (see Table 1). For instance, `llama-2-7b-chat-hf` gains 1.53% on MMLU, 1.34% on MBPP (pass@1), and 0.55% on GSM8K; `Gemma-7b` improves by 2.91% on MBPP (pass@10) and 1.67% on MathQA. Most gains lie between 0.5%–2.5%, peaking at 4.9%. As shown in Figure 3, DENOISE yields more concentrated improvements, whereas MODE exhibits greater variance—likely due to its routing mechanism.

**Out-of-Domain Performance.** Across WinoGrande and ARC-C, MODE consistently improves out-of-distribution reasoning. On all model scales, MODE yields 1–3 point gains over the baseline (e.g., +1.23% on WinoGrande and +1.68% on ARC-C for `LLaMA-2-7B`), with similar trends observed in larger LLaMA-2 and Gemma models. These results indicate that mitigating noise weights enhances robustness beyond in-domain tasks.

Table 2: DENOISE Performance of Different $p$ in $l_p$-norm on datasets.

|  | $p$ | MMLU | GSM8K | MathQA |
|---|---|---|---|---|
| acc(%) | 1 | 46.26 | 21.34 | 26.13 |
|  | 2 | 47.83 | 22.21 | 27.34 |
|  | 4 | 47.27 | 22.05 | 26.57 |

Table 3: DENOISE Performance of Different Denoising Ratio on MMLU.

|  | Ratio | MMLU | Ratio | MMLU |
|---|---|---|---|---|
| acc(%) | base | 45.81 | 5% | 47.79 |
|  | 1% | 47.75 | 10% | 47.83 |
|  | 3% | 47.83 | 20% | 45.33 |

Table 4: DENOISE Performance of Denoising Different Sublayers.

|  | Patch | GSM8K | MathQA |
|---|---|---|---|
| acc(%) | base | 20.24 | 25.33 |
|  | MLP | 21.61 | 26.87 |
|  | Self-Attention | 23.42 | 25.90 |
|  | Combination | 22.21 | 27.34 |

Table 5: MODE Performance of Different K-means on MMLU.

|  | K | MMLU | K | MMLU |
|---|---|---|---|---|
| acc(%) | 4 | 47.12 | 12 | 47.79 |
|  | 6 | 47.27 | 14 | 47.27 |
|  | 8 | 47.50 | 16 | 47.65 |
|  | 10 | 47.46 | 18 | 47.58 |

## 4.3 ABLATION STUDIES

In the ablation studies, we use the `llama-2-7b-chat-hf` model due to its stable performance in previous results. Employing DENOISE, we analyze $l_p$-norm, denoise ratios, layers, and patch-square configurations. Furthermore, employing MODE, we analyze the k-means clustering process.

$l_p$**-norm.** We firstly evaluate the performance of different $p$-norm values on the MMLU dataset. For $p = 2$, accuracy improves to 47.83%, representing a 2.02% increase over the base accuracy. For $p = 4$, accuracy decreases to 47.27%, noting that different $p$-norm configurations yield varying results in terms of performance, with $p = 2$ achieving the highest accuracy, as shown in Table 2.

**Denoising Ratio.** We denoise the all model layers (layers 0 to 31) using a $1 \times 1$ patch-square configuration and examine different ratios on the MMLU dataset grouped into 12 clusters. With a 10% denoising ratio, accuracy reaches 47.83%, a 2.02% increase over the 45.81% of base. This shows the 10% ratio effectively balances accuracy and computational cost, as shown in Table 3.

**Denoising Sublayers.** We then apply a 5% denoising to MLP layers and Self-Attention layers, from layers 0 to 31, evaluating the impact on the GSM8K and MathQA, grouped into 8 clusters. Results shows denoising the Self-Attention layers yields the best on GSM8K, with a 3.18% improvement. On MathQA, the combination performs best, reaching a 2.01% increase, as shown in Table 4.

**K-means Clustering.** Finally, we examine the impact of different K values applied in MODE in the K-means clustering step on the MMLU dataset, grouped into 12 clusters. We vary the number of clusters from 6 to 16, the results show that with **K=12**, the accuracy reaches the highest value of 47.79%, outperforming other cluster settings, as shown in Table 5.

**Patch-square Configuration.** Besides, based on 10% denoising ratio, we evaluate different patch-square configurations on the MBPP and HumanEval, grouped into 3 clusters. For MBPP, the $1 \times 1$ **patch-square** achieves the best performance on pass@1 with a 1.56% improvement, and on pass@10 with a 2.40% increase. Similarly, on HumanEval, the $1 \times 1$ configuration lead with 1.28% gains for pass@1 and 1.22% for pass@10 respectively over the baseline, as shown in Table 6.

Table 6: DENOISE Performance of Different Patch Size on MBPP and HumanEval.

|  | Patch | MBPP | | HumanEval | |
|---|---|---|---|---|---|
|  |  | @1(%) | @10(%) | @1(%) | @10(%) |
| pass@k | base | 19.24 | 23.60 | 14.45 | 19.51 |
|  | $1 \times 1$ | 20.80 | 26.00 | 15.73 | 20.73 |
|  | $2 \times 2$ | 19.88 | 25.60 | 15.67 | 20.12 |
|  | $4 \times 4$ | 18.58 | 24.00 | 14.88 | 20.73 |
|  | $16 \times 16$ | 18.40 | 24.40 | 13.97 | 17.68 |

## 5 RELATED WORKS

**Pruning.** Based on granularity, neural network pruning methods are categorized as unstructured, structured, or semi-structured. Unstructured pruning removes individual low-importance weights to achieve high sparsity; structured pruning eliminates entire structures such as neurons or channels (Blalock et al., 2020; Frantar & Alistarh, 2023a; Wang et al., 2019); semi-structured pruning offers a trade-off between flexibility and regularity (Syed et al., 2023). One of the core components of pruning is the design of importance metrics, which assess parameter relevance using heuristics such as magnitude, gradient-based estimates, or loss sensitivity (Han et al., 2015a; Molchanov et al., 2019; Sun et al., 2024). These metrics guide the removal of redundant parameters but have shown limited consistency and robustness across architectures and tasks (Sutskever et al., 2013). Our work reframes *pruning as a mechanism not only for structural simplification but also for enhancing model effectiveness*, providing a novel extension to its theoretical and practical scope.

**Mixture of Experts.** Most existing Mixture-of-Experts (MoE) architectures construct experts by replicating feed-forward networks (FFNs) within Transformer blocks, assuming specialization emerges implicitly through routing during training (Lepikhin et al., 2020; Fedus et al., 2022; Du et al., 2022). Recent works like DS-MoE (Pan et al., 2023) and the Emergent Modularity hypothesis (Zhang et al., 2023) provide empirical evidence of latent sparsity and modularity in FFN layers, while MoEfication (Zhang et al., 2022) and LLaMA-MoE (Zhu et al., 2024) further explore expert configuration through parameter reallocation and reuse without retraining. ToMoE (Gao et al., 2025) uses dynamic structural pruning to convert a dense LLM into an MoE model by training only the router and auxiliary modules. Rather than relying on implicit specialization or structural sparsification, our work deterministically identify and purify domain-relevant subnetwork experts, yielding a *tuning-free MoE that is adaptively aligned to the task domain*.

## 6 CONCLUSION

In this work, we empirically demonstrate that removing certain weights can unexpectedly improve the performance of LLMs on specific tasks. Building on this observation, we propose the DENoise algorithm, which prunes domain-aware noise weights from both attention and FFN layers to construct task-specialized experts, and further introduce the lightweight MoDE framework that can be seamlessly integrated into decoder-only LLM architectures. While these findings provide practical and deployable methods, our validation primarily relies on the $l_p$-norm to estimate weight importance; incorporating alternative metrics such as loss sensitivity, gradient-based scores, or influence functions could further strengthen our conclusions. Moreover, although the proposed perspective suggests the possible existence of noise neurons, our justification remains largely empirical, and we do not yet provide a generalized theoretical metric to precisely quantify them. Future work will therefore explore refined criteria, grounded in both empirical evidence and theory, to localize such harmful parameters more effectively, as well as extending the framework to diverse tasks such as open-ended generation, multi-modal reasoning, and real-world dialogue systems.

**Ethics Statement.** Our work concerns decoding efficiency in pre-trained models and involves no new human data. All benchmarks are publicly available and used under license. Methods modify inference only, without affecting training data or parameters, and introduce no additional data risks. Potential misuse is comparable to baseline models; responsible use is advised.

**Reproducibility Statement.** We document model versions, datasets, preprocessing, decoding hyperparameters, verifier rules, and software environment. Code and instructions will be released to ensure faithful replication.

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

## A    COMPLIANCE AND ADDITIONAL STATEMENTS

### A.1    USAGE OF LARGE LANGUAGE MODELS

This paper was written through the authors' independent reasoning and manual drafting. LLMs were not used for core tasks such as experiment design or analysis; at most, minor language refinement tools were consulted. Final decisions and revisions are entirely the authors'. Disclosure: *"No, not at all."*

### A.2    ETHICS STATEMENT

Our work concerns decoding efficiency in pre-trained models and involves no new human data. All benchmarks are publicly available and used under license. Methods modify inference only, without affecting training data or parameters, and introduce no additional data risks. Potential misuse is comparable to baseline models; responsible use is advised.

### A.3    REPRODUCIBILITY STATEMENT

We document model versions, datasets, preprocessing, decoding hyperparameters, verifier rules, and software environment. Code and instructions will be released to ensure faithful replication.

## B  SCALING HIDDEN LAYERS INTO PATCHES MAINTAINS INFORMATION IN SELF-ATTENTION AND MLP

In Transformer models, the Self-Attention and MLP layers are critical for capturing global contextual information and performing non-linear transformations on feature representations. The scaling of hidden layers into patches might raise concerns about the potential loss of information, but this process is designed to preserve both local and global relationships in the model.

**Global Context Preservation in Self-Attention.** The Self-Attention mechanism ensures that every token in the input sequence can attend to all other tokens, capturing global dependencies. This operation is described by:

$$\text{Attention}(Q, K, V) = \text{softmax}\left(\frac{QK^T}{\sqrt{d_k}}\right) V \tag{17}$$

Where: $Q$, $K$, and $V$ are the Query, Key, and Value matrices. $d_k$ is the dimensionality of the Key vectors. By scaling hidden layers into patches, each patch retains local interactions within the patch. Meanwhile, the Self-Attention mechanism ensures that global interactions between patches are maintained. This is because the attention mechanism operates across all patches, allowing the model to propagate global information and maintain context across the entire sequence of patches. As a result, the scaled matrix retains the full global context, ensuring no information is lost during patch scaling.

**Patch Scaling and Information Compression.** When the hidden layers are scaled by a factor $\alpha$, the resulting reduced matrix of size $\beta X \times \beta Y$ ($\beta = 1/\alpha$) has elements that correspond to patches in the original matrix. Each patch $P_{ij}$ captures a compressed representation of the information within the original matrix. By aggregating the contributions from each element in a patch, the scaled matrix effectively compresses the local information, while Self-Attention ensures that this compressed representation continues to interact globally. The importance of each patch is calculated as:

$$\theta_{ij} = \sum_{(m,n) \in P_{ij}} \left( |W_{mn}| \times \|X_{mn}\|_2 \right) \tag{18}$$

This compression allows for efficient representation of both local and global information, preserving the integrity of the original model.

**MLP Layer and Information Flow.** Following Self-Attention, the MLP layer processes the globally-contextualized output. The MLP is defined as:

$$\text{MLP}(h) = \sigma(W_2 \cdot \text{ReLU}(W_1 \cdot h)) \tag{19}$$

Where: $h$ is the output from Self-Attention. $W_1$ and $W_2$ are the weight matrices in the MLP. $\sigma$ is the activation function (typically ReLU). The MLP performs non-linear transformations on the compressed feature representations from the patches. Since the MLP does not rely on spatial relationships, it processes the patch-level information without any risk of information loss. The critical feature transformations in the MLP are unaffected by the scaling process, ensuring that the information flow remains intact.

## C  PROCEDURE OF DENOISE TO SELECT BEST $\theta$

In this section, we present the procedure for selecting the optimal threshold $\theta$ in the DENoise method. The goal is to assess the impact of varying $\theta$ values on performance across multiple datasets and domains, specifically MMLU, GSM8K, MathQA, and HumanEval. Each table provides a comprehensive comparison of the DENoise performance over different ranges of $\theta$, from 50% to 100%, highlighting its effectiveness in selecting the most relevant experts in various domains.

# D THRESHOLD-BASED PERFORMANCE ANALYSIS ACROSS DATASETS

This appendix provides a comprehensive analysis of the performance trends observed across varying threshold $\theta$ values for the datasets GSM8K, MathQA, HumanEval, MBPP, and MMLU. Each dataset, representing a distinct domain, showcases unique response patterns when applying the MoDE framework. As $\theta$ increases, we observe noticeable fluctuations in accuracy, highlighting the dynamic behavior of domain-specific subnetworks. The results consistently demonstrate that activating experts based on denoised domain-specific weights yields stable improvements across tasks. This analysis reinforces the scalability and adaptability of MoDE, validating its ability to enhance task-specific accuracy without the need for fine-tuning.

Table 7: Performance comparison of DENOISE throughout all MMLU domains with different $\theta$.

| cluster_id | Samples | 50-55% | 55-60% | 60-65% | 65-70% | 70-75% | 75-80% | 80-85% | 85-90% | 90-95% | 95-100% | Max (%) | Ratio (%) |
|---|---|---|---|---|---|---|---|---|---|---|---|---|---|
| mmlu_0 | 2047 | 57.79 | 58.72 | 58.96 | 59.26 | 59.99 | 59.94 | 60.67 | 60.23 | 60.77 | 60.92 | 60.92 | 14.58 |
| mmlu_1 | 1518 | 35.57 | 34.72 | 34.32 | 35.70 | 35.38 | 35.31 | 35.84 | 35.44 | 36.17 | 35.77 | 36.17 | 10.81 |
| mmlu_2 | 895 | 23.02 | 24.02 | 22.57 | 23.35 | 23.02 | 22.01 | 22.46 | 22.68 | 22.79 | 22.57 | 24.02 | 6.37 |
| mmlu_3 | 1586 | 48.93 | 47.92 | 49.37 | 48.99 | 49.87 | 49.50 | 49.43 | 50.44 | 50.88 | 51.01 | 51.01 | 11.29 |
| mmlu_4 | 212 | 27.83 | 28.77 | 28.30 | 25.94 | 28.77 | 26.89 | 27.83 | 26.89 | 27.36 | 26.89 | 28.77 | 1.51 |
| mmlu_5 | 1477 | 59.58 | 59.04 | 60.39 | 59.51 | 60.12 | 60.80 | 60.93 | 61.61 | 61.61 | 61.75 | 61.75 | 10.52 |
| mmlu_6 | 322 | 35.09 | 32.92 | 33.85 | 34.78 | 33.54 | 34.78 | 33.54 | 34.16 | 35.09 | 35.09 | 35.09 | 2.29 |
| mmlu_7 | 434 | 27.65 | 25.58 | 29.95 | 26.96 | 26.96 | 27.19 | 28.11 | 29.72 | 27.19 | 29.49 | 29.95 | 3.09 |
| mmlu_8 | 2016 | 55.21 | 55.36 | 55.46 | 55.51 | 56.15 | 55.56 | 56.35 | 57.04 | 57.19 | 57.44 | 57.44 | 14.36 |
| mmlu_9 | 1839 | 46.66 | 47.53 | 46.82 | 46.49 | 47.36 | 47.74 | 47.36 | 48.02 | 48.45 | 48.29 | 48.45 | 13.09 |
| mmlu_10 | 1174 | 30.92 | 30.15 | 31.26 | 31.09 | 30.49 | 30.15 | 30.83 | 32.03 | 32.28 | 31.86 | 32.28 | 8.36 |
| mmlu_11 | 522 | 42.34 | 45.21 | 44.25 | 42.91 | 46.74 | 45.40 | 46.74 | 47.32 | 46.36 | 47.13 | 47.32 | 3.72 |

Table 8: Performance comparison of DENOISE throughout all GSM8K domains with different $\theta$.

| cluster_id | Samples | 50-55% | 55-60% | 60-65% | 65-70% | 70-75% | 75-80% | 80-85% | 85-90% | 90-95% | 95-100% | Max (%) | Ratio (%) |
|---|---|---|---|---|---|---|---|---|---|---|---|---|---|
| gsm8k_0 | 240 | 12.92 | 15.00 | 15.83 | 18.75 | 13.75 | 17.08 | 16.67 | 17.50 | 15.83 | 16.67 | 18.75 | 18.20 |
| gsm8k_1 | 8 | 0.00 | 12.50 | 12.50 | 37.50 | 25.00 | 12.50 | 12.50 | 37.50 | 0.00 | 12.50 | 37.50 | 0.61 |
| gsm8k_2 | 225 | 17.78 | 21.78 | 18.22 | 24.00 | 22.67 | 22.67 | 22.22 | 21.78 | 20.44 | 24.00 | 24.00 | 17.06 |
| gsm8k_3 | 361 | 18.28 | 16.90 | 19.67 | 20.50 | 19.94 | 23.82 | 21.05 | 23.82 | 21.61 | 23.82 | 23.82 | 27.37 |
| gsm8k_4 | 113 | 13.27 | 17.70 | 9.73 | 15.04 | 15.93 | 19.47 | 17.70 | 13.27 | 17.70 | 16.81 | 19.47 | 8.57 |
| gsm8k_5 | 193 | 12.44 | 10.88 | 12.95 | 13.99 | 15.03 | 17.62 | 20.73 | 18.65 | 17.10 | 19.69 | 20.73 | 14.63 |
| gsm8k_6 | 6 | 33.33 | 16.67 | 50.00 | 33.33 | 33.33 | 16.67 | 33.33 | 33.33 | 16.67 | 50.00 | 50.00 | 0.45 |
| gsm8k_7 | 173 | 16.18 | 14.45 | 17.92 | 23.12 | 17.34 | 19.08 | 17.34 | 19.65 | 18.50 | 19.65 | 23.12 | 13.12 |

Table 9: Performance comparison of DENOISE throughout all MathQA domains with different $\theta$.

| cluster_id | Samples | 50-55% | 55-60% | 60-65% | 65-70% | 70-75% | 75-80% | 80-85% | 85-90% | 90-95% | 95-100% | Max (%) | Ratio (%) |
|---|---|---|---|---|---|---|---|---|---|---|---|---|---|
| mathqa_0 | 289 | 19.03 | 19.72 | 20.42 | 23.53 | 26.99 | 28.72 | 26.30 | 22.84 | 26.30 | 22.15 | 28.72 | 9.68 |
| mathqa_1 | 318 | 24.53 | 24.84 | 24.21 | 22.96 | 31.45 | 23.58 | 29.87 | 29.25 | 26.42 | 25.79 | 31.45 | 10.65 |
| mathqa_2 | 453 | 20.75 | 22.30 | 27.15 | 22.96 | 24.06 | 25.39 | 23.18 | 26.49 | 25.39 | 22.96 | 27.15 | 15.18 |
| mathqa_3 | 107 | 24.30 | 27.10 | 28.97 | 20.56 | 28.04 | 27.10 | 28.04 | 20.56 | 22.43 | 20.56 | 28.97 | 3.58 |
| mathqa_4 | 238 | 24.37 | 20.17 | 29.83 | 21.01 | 22.69 | 29.41 | 22.69 | 27.31 | 29.41 | 28.15 | 29.83 | 7.97 |
| mathqa_5 | 269 | 26.77 | 20.45 | 24.91 | 25.65 | 29.37 | 27.14 | 25.65 | 21.56 | 23.79 | 29.00 | 29.37 | 9.01 |
| mathqa_6 | 659 | 24.28 | 19.58 | 20.49 | 21.40 | 20.64 | 22.91 | 21.55 | 18.97 | 20.64 | 19.88 | 24.28 | 22.08 |
| mathqa_7 | 652 | 20.09 | 21.47 | 21.32 | 24.08 | 22.70 | 22.70 | 25.92 | 24.54 | 23.47 | 22.55 | 25.92 | 21.84 |

Table 10: Performance comparison of DENOISE throughout all HumanEval domains with different $\theta$.

| cluster_id | Metric | Samples | 50-55% | 55-60% | 60-65% | 65-70% | 70-75% | 75-80% | 80-85% | 85-90% | 90-95% | 95-100% | Max (%) | Ratio (%) |
|---|---|---|---|---|---|---|---|---|---|---|---|---|---|---|
| humaneval_0 | pass@1 | 44 | 11.82 | 17.05 | 15.68 | 14.77 | 15.00 | 16.36 | 17.05 | 15.23 | 15.45 | 14.77 | 17.05 | 26.83 |
| humaneval_1 | pass@1 | 75 | 8.27 | 9.47 | 10.80 | 10.67 | 13.07 | 15.33 | 12.67 | 13.20 | 13.47 | 13.33 | 15.33 | 45.73 |
| humaneval_2 | pass@1 | 45 | 6.89 | 12.22 | 11.11 | 9.33 | 14.22 | 14.00 | 15.11 | 14.00 | 14.89 | 15.11 | 15.11 | 27.44 |
| humaneval_0 | pass@10 | 44 | 18.18 | 25.00 | 25.00 | 18.18 | 22.73 | 18.18 | 25.00 | 18.18 | 20.45 | 20.45 | 25.00 | 26.83 |
| humaneval_1 | pass@10 | 75 | 10.67 | 14.67 | 13.33 | 12.00 | 18.67 | 18.67 | 16.00 | 17.33 | 17.33 | 16.00 | 18.67 | 45.73 |
| humaneval_2 | pass@10 | 45 | 8.89 | 15.56 | 15.56 | 13.33 | 15.56 | 15.56 | 17.78 | 15.56 | 20.00 | 17.78 | 20.00 | 27.44 |

Table 11: Performance comparison of DENOISE throughout all MBPP domains with different $\theta$.

| cluster_id | Metric | Samples | 50-55% | 55-60% | 60-65% | 65-70% | 70-75% | 75-80% | 80-85% | 85-90% | 90-95% | 95-100% | Max (%) | Ratio (%) |
|---|---|---|---|---|---|---|---|---|---|---|---|---|---|---|
| mbpp_0 | pass@1 | 185 | 35.68 | 34.32 | 33.89 | 38.16 | 34.05 | 35.19 | 37.51 | 34.65 | 36.65 | 36.38 | 38.16 | 37.00 |
| mbpp_1 | pass@1 | 53 | 17.74 | 15.47 | 20.19 | 27.92 | 22.83 | 25.47 | 23.96 | 20.75 | 19.62 | 22.08 | 27.92 | 10.60 |
| mbpp_2 | pass@1 | 262 | 7.29 | 6.18 | 5.84 | 6.34 | 6.56 | 8.09 | 6.56 | 6.45 | 7.75 | 7.52 | 8.09 | 52.40 |
| mbpp_0 | pass@10 | 185 | 40.54 | 43.24 | 42.16 | 42.16 | 40.54 | 41.08 | 44.32 | 39.46 | 42.70 | 42.16 | 44.32 | 37.00 |
| mbpp_1 | pass@10 | 53 | 24.53 | 26.42 | 28.30 | 32.08 | 28.30 | 35.85 | 30.19 | 26.42 | 26.42 | 33.96 | 35.85 | 10.60 |
| mbpp_2 | pass@10 | 262 | 9.16 | 9.16 | 9.92 | 9.16 | 10.31 | 11.83 | 9.92 | 10.31 | 11.83 | 11.45 | 11.83 | 52.40 |

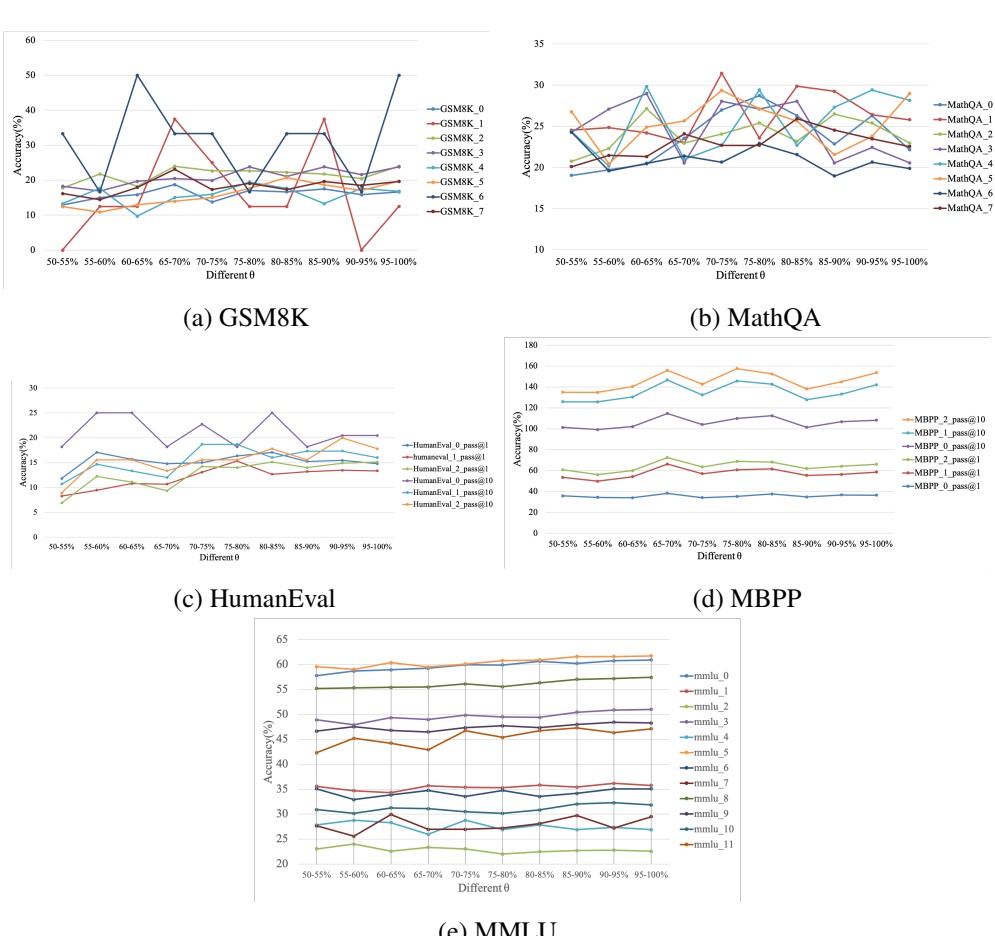

(a) GSM8K

(b) MathQA

(c) HumanEval

(d) MBPP

(e) MMLU

Figure 4: Accuracy comparison of DENOISE across different thresholds $\theta$ for various datasets including GSM8K, MathQA, HumanEval, MBPP, and MMLU. Each subfigure (a-e) shows performance variations with respect to the $\theta$ values, highlighting dataset-specific accuracy trends.

# E DETAILED DERIVATION OF DOMAIN-INDUCED IMPORTANCE SHIFTS

## E.1 PRELIMINARIES

Consider a pretrained network with parameters $\mathbf{W} = \{W_i\}$ and an input distribution $\mathcal{D}$. For a small perturbation $\Delta W_i$, the first-order Taylor expansion of the loss gives

$$\Delta \mathcal{L} \approx \sum_i g_i \, \Delta W_i, \tag{20}$$

where $g_i = \frac{\partial \mathcal{L}}{\partial W_i}$ is the gradient w.r.t. weight $W_i$. Under magnitude-based or norm-based pruning, the implicit assumption is that weights with small $|W_i|$ have small contributions to the loss and thus can be safely removed.

However, this perspective treats the importance of $W_i$ as *domain-invariant*, ignoring that the gradient $g_i$ is an expectation over the underlying data distribution. When the domain shifts from the pretraining distribution $\mathcal{D}_{\text{pt}}$ to a downstream domain $\mathcal{D}_{\text{in}}$, the gradient changes accordingly:

$$g_i^{(\text{in})} = \mathbb{E}_{(x,y) \sim \mathcal{D}_{\text{in}}} \left[ \frac{\partial \ell(f(x; \mathbf{W}), y)}{\partial W_i} \right]. \tag{21}$$

## E.2 DOMAIN-INDUCED REVERSAL OF WEIGHT CONTRIBUTION

Define the global importance as

$$I^{(\text{pt})}(i) = -g_i^{(\text{pt})} W_i, \tag{22}$$

which approximates the marginal loss change if weight $W_i$ is zeroed out.

Under domain shift, this becomes

$$I^{(\text{in})}(i) = -g_i^{(\text{in})} W_i. \tag{23}$$

A weight becomes a *noise weight* (i.e., harmful in-domain) when

$$I^{(\text{in})}(i) > 0 \quad \text{but} \quad I^{(\text{pt})}(i) \le 0, \tag{24}$$

meaning that the sign of the gradient flips between domains. This typically occurs when the downstream distribution induces responses that conflict with the behavior learned during pretraining.

## E.3 WHY MID-RANKED WEIGHTS ARE MOST SENSITIVE

Magnitude-based proxies such as $|W_i| \cdot \|X_i\|_2$ correlate with the weight–activation interaction term in the Taylor expansion:

$$g_i \approx \mathbb{E}[X_i^\top \delta], \tag{25}$$

where $X_i$ is the input to weight $W_i$ and $\delta$ is the backpropagated error.

Large-$|W_i|$ weights dominate the loss landscape and remain stable under domain shift, so they rarely flip from beneficial to harmful. Very small-$|W_i|$ weights have negligible contribution in either domain.

The weights most sensitive to sign reversal satisfy:

$$|W_i| \text{ moderate}, \quad \|X_i\|_2 \text{ moderate}, \tag{26}$$

leading to a small but non-negligible $g_i W_i$ term whose sign is easily perturbed by domain-induced gradient changes.

Thus, domain-induced noise weights $N_D$ are most likely to appear in the *mid-ranked region* of global importance.

### E.4 DOMAIN-AWARE PROXY

Given the correlation between $|W_i| \cdot \|X_i\|_2$ and $I^{(pt)}$, we use this quantity as a global proxy to identify the region where domain-sensitive weights concentrate.

We therefore define the approximate noise-weight search region as

$$R = \{i \mid |W_i| \cdot \|X_i\|_2 \in [p_{20}, p_{70}]\}, \tag{27}$$

where $p_{20}$ and $p_{70}$ are empirical quantiles. Within $R$, we compute domain-specific importance:

$$I_D(i) = -g_i^{(D)} W_i, \tag{28}$$

and classify noise weights as

$$N_D = \{i \in R \mid I_D(i) > 0\}. \tag{29}$$

## F FISHER INFORMATION PERSPECTIVE ON NOISE WEIGHTS.

To provide a theoretical foundation for the Noise Weight, we draw upon statistical learning theory. The importance of a parameter can be evaluated not only by its magnitude but also by its sensitivity to the loss function. This sensitivity is formally captured by the Fisher Information Matrix (FIM), defined for a parameter $\theta$ as:

$$\mathcal{I}(\theta) = \mathbb{E}_{x \sim \mathcal{D}} \left[ \left( \frac{\partial}{\partial \theta} \log p(y|x; \theta) \right)^2 \right], \tag{30}$$

where $\mathcal{D}$ denotes the task-specific data distribution. Parameters with low Fisher information contribute negligibly to the curvature of the loss landscape, implying weak or unstable influence on predictions. Such parameters can be regarded as *noise weights*, since their activation may increase variance without providing meaningful task-specific signal.

Moreover, under the PAC-Bayes framework, the generalization error of a model is bounded by:

$$\mathbb{E}_{\theta \sim Q}[L(\theta)] \leq \hat{L}(\theta) + \sqrt{\frac{KL(Q \| P) + \ln \frac{2\sqrt{n}}{\delta}}{2n}}, \tag{31}$$

where $Q$ is the posterior distribution over parameters, $P$ is a prior, $n$ is the sample size, and $\delta$ is a confidence parameter. The inclusion of noise weights effectively enlarges the support of $Q$, increasing the KL divergence term and thereby loosening the bound. Conversely, removing noise weights can be seen as constraining the hypothesis space, reducing the model complexity, and tightening the generalization bound. This perspective may explain why pruning certain mid-to-low ranked weights—although counterintuitive under classical importance metrics—may actually *improve* in-domain performance.

