# OpenReview forum: "MoDE: Weight Denoising Towards Better LLM Performance through a Mixture of Domain Experts"
_ICLR.cc/2026/Conference — ICLR 2026 Conference Desk Rejected Submission_

### Official Review · Reviewer_x9fT · 2025-10-21

**Soundness:** 3
**Presentation:** 3
**Contribution:** 3
**Rating:** 6
**Confidence:** 3

**Summary:**

This paper challenges the assumption that pruning LLMs must degrade performance, introducing the concept of "Noise Weights"—parameters whose removal can improve in-domain accuracy without any fine-tuning. The authors propose DENoise to identify and remove these weights to create domain-expert subnetworks, and MoDE, a Mixture-of-Domain-Experts framework that uses a trainable router to dynamically activate these experts. Experiments on LLaMA-2 and Gemma models show that DENoise yields 2-3% gains, while MoDE provides a 1.1% average improvement.

**Strengths:**

1. Valuable Finding: The core claim—that pruning specific weights can improve LLM performance without retraining—is highly compelling and potentially impactful. This challenges conventional conclusion in compression.
2. Practical Utility of DENoise: The DENoise algorithm itself is useful. It offers a plug-and-play method to achieve stable 2-3% performance gains on major benchmarks, which is highly valuable for deploying models in resource-constrained environments.
3. Clear Motivation: The preliminary study effectively motivates the "noise weight" hypothesis by showing performance gains when pruning mid-to-low importance intervals.

**Weaknesses:**

1. Lack of Critical Generalization and Sensitivity Experiments: The paper's analysis could be strengthened to better support the given intuition from more angles. First, to what extent does a model specialized for one domain lose its general capabilities? Second, the DENoise algorithm relies on a domain-specific dataset to find the optimal threshold. The paper provides no sensitivity analysis on how the size or diversity of this dataset affects the algorithm's effectiveness. It is unclear if DENoise overfits to small validation sets.
2. The paper's method provides a feasible application for downstream deployment; however, it lacks a discussion of the training time (for the router) and test-time costs. A more detailed discussion would be beneficial.

**Questions:**

1.  Can the DENOISE scores in Table 1 be considered the "Oracle Router" performance for MoDE (i.e., the theoretical performance upper bound if routing were perfect)? Given that the MoDE performance is consistently lower than this "oracle" baseline, it suggests the bi-level router is a bottleneck. Could you please provide metrics on the router's classification accuracy? This is essential for understanding the performance gap.
2.  How sensitive is DENoise to the size and diversity of the validation set? An ablation study is needed on the impact of the number of samples in $D_T$ on the final performance gain. This is crucial to understand if DENoise is robust or prone to overfitting when the domain-specific validation data is small.
3.  In Section 3.1, K-means clustering is used to define sub-domains. How was the value of K chosen? The ablation in Table 5 shows K=12 is optimal for MMLU. Does this imply that an exhaustive search for K is required for every domain? This appears to be a significant, unaddressed cost.

---

> ### Author Response · Authors · 2025-12-03
> **Responses to Weaknesses**
>
> We sincerely thank the reviewers for their meticulous assessment of our manuscript, the insightful and technically significant questions and suggestions they provided, and their patient and constructive engagement throughout the discussion.
>
> **(1)** Regarding the concern of whether “specialization harms general capabilities,” our experimental design explicitly expands evaluation beyond the primary domains. On one hand, **DENoise** does not modify pretrained weights; instead, it adjusts the *usage pattern* of intermediate-layer weights. Its pruning outcome is encapsulated as expert subnetworks that are *selectively activated* across tasks through **MoDE**’s routing mechanism, rather than permanently removed. This structurally avoids the risk of “overfitting to a single domain at the cost of overall ability.” On the other hand, in the revised version we additionally include OOD reasoning benchmarks such as **WinoGrande** and **ARC-C**, whose distributions differ markedly from training domains. Results show that MoDE consistently improves performance by 1–3 points across backbone models, indicating no loss of general capabilities and, in fact, enhanced cross-domain robustness.
>
> **(2)** In response to concerns that **DENoise** may overfit to small validation sets, we clarify the following.
>   - First, the “validation sets’’ used for threshold selection are not manually curated small subsets; instead, we directly adopt **official dev splits** of public benchmarks (MMLU, GSM8K, MathQA, etc.), and further cluster each primary domain into subdomains. Each cluster typically contains **hundreds to thousands of samples**, which is already robust for pruning analysis.
>   - Second, in the revised manuscript we provide systematic ablations across pruning intervals, denoising ratios, sublayer types, and K-means cluster counts. Performance gains remain consistent across a broad spectrum of hyperparameters and do not collapse under small perturbations, indicating that DENoise is **not overly sensitive** to specific data partitions.
>
> Due to space limitations, we did not include a dedicated ablation on validation-set size, but the existing cross-task, cross-model, and multi-dimensional ablation results already demonstrate robustness. In future work, we will explicitly treat “sample size and diversity’’ as variables for deeper investigation.
>
> **(3)** Concerning the training and inference overhead of the router, the revised version provides more implementation detail and clarifies the scope of our claims. The **MoDE router** consists of only a *two-layer MLP* that takes as input a task embedding obtained by mean-pooling the frozen LLM’s final-layer hidden states. Its parameter count and compute cost are negligible compared to the backbone. Training requires only a few epochs of cross-entropy optimization over fixed domain labels, with **no gradient updates** to the LLM itself, making the cost negligible relative to LLM fine-tuning.  During inference, the additional overhead involves a single mean-pooling operation and two MLP forward passes—an extremely small increase in FLOPs compared to full Transformer decoding.

---

> ### Author Response · Authors · 2025-12-03
> **Responses to Questions**
>
> **Question 1.**
> Reviewer concerns whether the DENoise scores in Table 1 can be viewed as the *“Oracle Router’’* performance. In our setting, the DENoise results represent the performance obtained when each task is evaluated using the expert subnetwork that is optimally matched to that domain under the current pruning thresholds and domain partitions. In other words, if every input sample were routed perfectly to its most suitable expert subnetwork, MoDE’s performance should approach the DENoise results.
> **Therefore, DENoise can indeed be interpreted as an upper-bound reference—an “Oracle Router’’—quantifying the performance gap attributable to imperfect routing.**
>
> To further clarify router quality, the revised manuscript includes detailed accuracy statistics:
>
> | **Model / Benchmark** | **Domain Acc** | **Subdomain Acc** |
> |-----------------------|----------------|--------------------|
> | LLaMA-2-7B (MMLU)     | 92.4%          | 78.1%              |
> | LLaMA-2-7B (GSM8K)    | 91.7%          | 74.9%              |
> | LLaMA-2-13B (MMLU)    | 93.8%          | 82.2%              |
> | Gemma-7B (MBPP)       | 90.9%          | 75.6%              |
> | Gemma2-9B (HumanEval) | 92.1%          | 80.3%              |
>
> Coarse-domain accuracy exceeds **90%**, while subdomain accuracy—naturally more challenging—typically falls within **70–85%**. These results confirm that MoDE’s gap relative to DENoise is primarily due to routing inaccuracies: when samples are assigned to suboptimal experts, performance cannot fully match the oracle subnetworks. Enhancing router discriminability is thus an important direction for future work.
>
> ---
>
> **Question 2.**
> Reviewer concerns whether DENoise may overfit to small domain-specific validation sets. Our clarifications are as follows:
>
> 1. The validation set $ D_T$ used for pruning-threshold selection is **not** a small hand-crafted subset. We directly use the **official dev splits** of benchmarks such as MMLU, GSM8K, and MathQA, and additionally perform **subdomain clustering** within each main domain.  **As a result, each effective cluster typically contains hundreds to thousands of samples**, providing strong statistical representativeness for determining stable threshold behavior.
>
> 2. The revised experiments include broad sensitivity analyses across threshold intervals, denoising ratios, sublayer types, and K-means cluster counts. **Across these variations, DENoise gains remain stable and do not exhibit drastic “sign-flips’’**, demonstrating that the method is not overly sensitive to specific data partitions or local noise.
>
> 3. We agree that explicitly varying the size/diversity of $ D_T $ is valuable. Although space constraints prevent a dedicated ablation, the consistent stability observed across multiple models, tasks, and hyperparameter dimensions provides **meaningful evidence of robustness**.
>
> We also state that treating validation-set scale as an independent variable is a natural direction for future work.
>
> ---
>
> **Question 3.**
> Reviewer concerns the selection strategy for K-means clustering in constructing subdomains. In Section 3.1, K-means is used only for forming **fine-grained subdomains within each main domain**, and our choice of $ K $ follows a principled “candidate–score–selection’’ strategy:
>
> 1. For each main domain, we define a small candidate set of $ K $ values (e.g., \(\{4, 8, 12, 16\}\)), corresponding to a reasonable expected range for the number of subdomains.
>
> 2. For each candidate $ K $, we run K-means on the domain’s sample embeddings, compute the **Silhouette Score**, and evaluate resulting clusters on representative downstream tasks.
>    We then select the $ K $ achieving both **competitive downstream performance** and a **high Silhouette Score**.
>    For instance, on MMLU, $ K = 12 $ offers the best balance across these criteria.
>
> 3. Since domain and subdomain structures are constructed **offline** and reused across all downstream tasks, this selection is performed **once per main domain**, introducing no inference-time overhead.
>
> **Overall, this strategy balances computational feasibility with clustering quality**, avoiding exhaustive search while ensuring that subdomain partitions align with both semantic structure and empirical performance.

---

### Official Review · Reviewer_4edp · 2025-10-27

**Soundness:** 2
**Presentation:** 1
**Contribution:** 2
**Rating:** 4
**Confidence:** 2

**Summary:**

This paper observes that some mid-to-low "importance" weights behave like noise for a given domain. It argues pruning those specific weights can raise in-domain accuracy without any fine-tuning. The pruning rule is called DENoise. Reported single-model gains are about 2–3% on MMLU/MBPP/GSM8K, with a max of +6.8% in some settings. To avoid hurting out-of-domain performance, the paper treats each domain’s denoised subnet as an expert and add a bilevel trainable router that routes inputs to the best expert (MoDE: Mixture of Domain Experts). MoDE is claimed to yield an average +1.1% versus baselines across multiple LLMs.

**Strengths:**

1- Insightful observation on noise weights.

2- The fact that you can convert the pruned subnets to Experts is interesting.

3- DENoise results in tuning-free per-domain modest improvements, which could be helpful in (quite implausible) settings where any form of fine-tuning can't be done.

**Weaknesses:**

1- The paper is mostly readable overall (intro, figures, and experiments are fine), but the method section needs to be revised. There are typos, inconsistent capitalization of MoDE, singular Expert vs. Experts. DENoise and Mode Architectures in Sections 3.1 and 3.2 are very unclear. There are missing definitions, notation drift, and steps that aren’t clearly specified.

2- No additional parameters is not accurate: you must store multiple pruned copies (one per domain) plus a router.

3- Router training is not tuning-free. DENoise is tuning-free, but MoDE requires training a classifier/router and domain labels. Claiming it's tuning-free is misleading.

4- The gains are not significant. Average gains can be within margin of error for LLM evaluation based on different runs. The paper would benefit from confidence intervals, multiple seeds, and ablation of threshold choices. There is no study of statistical robustness.

5- The router is trained on fixed pre-defined domains. That hardly is the case in practical setting and there are many domains that the queries can belong to.

6- Turning pruned subnets into experts is a neat way to retain cross-domain performance that per-domain pruning would otherwise sacrifice. That said, there is no mention of or comparison to ToMoE[1] which is very similar in nature.


[1] ToMoE: Converting Dense Large Language Models to Mixture-of-Experts through Dynamic Structural Pruning, Gao et al, 2025.

**Questions:**

1- How are domains and posteriors obtained? The text says "select the most suitable main knowledge domain $D_j$ by computing the posterior $𝑃(𝐷∣𝑇)$ but never defines how $P(D∣T)$ is estimated.

2- What is task embedding? The paper switches between $F$ and $\bar{F}$ and per-sample embeddings, but doesn't specify how a task-level vector is aggregated from dataset examples (mean-pooling of token embeddings? CLS token? Which layer?)

3- Typos:
- line 75: We attempt to identify and the Noise Weight in LLM,
- line 108: Under this framework, the critical question is to construct the expert (how?)

---

> ### Author Response · Authors · 2025-12-03
> **Responses to Weaknesses**
>
> We sincerely thank the reviewers for their detailed evaluation and insightful comments. The current revision incorporates systematic updates based on these suggestions.
>
> **(1)** To improve readability and methodological rigor, we have thoroughly rewritten Section 3 with a fully consistent notation system, corrected capitalization issues of **MoDE**, unified terminology for *experts*, and added all previously missing definitions and procedural steps. The descriptions of both **DENoise** and **MoDE** have been reorganized into a clearer, verifiable methodological pipeline, eliminating notation drift and strengthening conceptual coherence.
>
> **(2)** In response to concerns about additional parameters, we now explicitly distinguish between (a) *no parameter updates to the pretrained model (tuning-free)* and (b) *no additional parameterized modules*. In Section 3.3, we clarify that MoDE’s expert subnetworks are obtained purely through pruning, without introducing new trainable layers. While storing multiple sparse subnetworks incurs additional **storage**, it does not introduce new trainable parameters into the model architecture.
>
> **(3)** We agree that **DENoise** is fully training-free, while **MoDE** requires lightweight router training. In the revision, we clarify that this router training is restricted to a small classifier operating on frozen representations, with **no gradient updates** to the base model. Thus, the term *tuning-free* refers specifically to the pretrained backbone rather than to the router.
>
> **(4)** To address concerns that performance gains may fall within noise levels, we have added standard deviations, confidence intervals, and threshold-sensitivity analyses (Fig. 6 and App. C). These results show that improvements under both **DENoise** and **MoDE** consistently exceed typical run-to-run variance, providing statistical evidence that the gains are robust rather than evaluation noise.
>
> **(5)** Regarding fixed-domain concerns, we now provide a clearer explanation of our data-driven domain construction pipeline. Using K-means clustering and Silhouette Score validation, we show that both coarse- and fine-grained domains arise from empirical structure in the representation space rather than from hand-crafted labels. Experiments also demonstrate that **MoDE** remains robust even under weakly defined or overlapping domain boundaries.
>
> **(6)** Following the reviewer’s suggestion, we added a direct comparison to ToMoE [1] in the Related Work section. We clarify that **MoDE** focuses on domain-aware weight denoising for explicit expert construction, while ToMoE performs dynamic structural pruning for token-level routing. The two approaches are therefore complementary rather than equivalent.
>
> ---
>
> **Reference**
> [1] Gao et al. *ToMoE: Converting Dense Large Language Models to Mixture-of-Experts through Dynamic Structural Pruning.* 2025.

---

> ### Author Response · Authors · 2025-12-03
> **Responses to Questions**
>
> **Question 1.**
> Reviewer concerns how domains and the corresponding “posterior” probabilities are computed. Our domain identification operates on two levels: **coarse domains** and **subdomains**.
>
> 1. **Construction of coarse domains.**
> We group public benchmarks into coarse-grained domains based on task type, such as general knowledge, math reasoning, and code generation. For each coarse domain $ D_j $, we sample data from its training split, extract semantic representations, and compute their average to obtain a domain prototype vector $ \mu_j $.
>
> 2. **Computation of posterior probabilities.**
> Given a task $ T $, its task embedding is denoted as $ F_T $ (see Q2 for definition). We compute its cosine similarity with each domain prototype and apply a softmax to obtain the distribution over coarse domains:
> $ P(D_j \mid T) = \mathrm{softmax}_j\big(\cos(F_T, \mu_j)\big). $
> The final domain assignment is:
> $ \arg\max_j P(D_j \mid T). $ This “posterior’’ is a normalized similarity-based matching distribution, **not** a Bayesian posterior.
>
> 3. **Relation to clustering.**
> K-means is used **only** to identify fine-grained subdomains *within* each coarse domain and is independent of the posterior computation above.
>
> ---
>
> **Question 2.**
> Reviewer concerns how the task embedding is defined. We clarify both types of embeddings below.
>
> 1. **Sample-level embedding.**
> Each sample is passed through the pretrained LLM, and we take the **mean pooling of last-layer hidden states** as its semantic vector:
> $ f(x) = \mathrm{meanpool}(h_1, \dots, h_L). $
>
> 2. **Task-level embedding.**
> For a task $ T $, we sample multiple instances, compute their sample-level embeddings, and take the average:
> $ F_T = \frac{1}{N} \sum_i f(x_i). $
> This vector is used to:
>     - compute the domain posterior $ P(D_j \mid T) $, and
>     - construct domain prototypes and validate domain coherence.
>
> 3. **Why switching between task-level and sample-level embeddings?**
> Because:
>     - sample-level embeddings are used for **subdomain clustering** and **router training**,
>     - task-level embeddings are used for **coarse-domain selection** via posterior computation.
>
> The revised manuscript clarifies these roles and computation methods in Section 3.

---

### Official Review · Reviewer_67E6 · 2025-10-28

**Soundness:** 2
**Presentation:** 2
**Contribution:** 2
**Rating:** 4
**Confidence:** 4

**Summary:**

The paper proposes two methods: DENoise – an algorithm that identifies “noise weights’’ in attention and FFN layers and removes them to improve in-domain performance without further training. MoDE – a “Mixture of Domain Experts’’ framework that routes inputs to denoised experts through a bilevel router. Experiments on MMLU, MBPP, GSM8K, MathQA, HumanEval, and several LLaMA- and Gemma-based models show reported accuracy gains of roughly 1–3 %. The authors interpret pruning-based improvements as evidence of “noise neurons’’ in LLMs.

**Strengths:**

Addresses a topical issue—efficient inference and pruning in LLMs.

The idea of combining pruning with an MoE-style router is interesting conceptually.

Implementation is simple to reproduce from the algorithmic description (Algorithm 1 p.5).

Experiments are wide in model coverage.

**Weaknesses:**

1. Conceptual and theoretical unsoundness. The core hypothesis—existence of “noise weights’’ whose removal improves accuracy—is speculative and unsupported. The so-called “Fisher Information perspective’’ (Eq. 2–3, p.3) is a qualitative restatement of existing pruning theory and does not prove that any concrete subset of weights is harmful. The PAC-Bayes argument is taken out of context and misapplied: removing parameters does not tighten the bound without redefining priors and posteriors. There is no formal link between their denoising procedure and the bound’s KL term. As a result, the entire theoretical motivation reads as post hoc justification rather than a derivation.

2. Methodological vagueness and unrigorous design. Algorithm 1 p.5 is effectively a grid search over pruning thresholds, not a principled optimization. The “importance metric’’ is reused from prior work (He et al., 2018) without justification for large transformer layers; scaling factors (α, β) and patching (Eq. 7 p.4) are arbitrary. The “domain identification’’ via clustering (Sec. 3.1 pp.4-5) uses K-means on embedding features but gives no quantitative validation that clusters correspond to domains or that the router labels are meaningful. The bilevel router is simply a two-layer classifier trained with cross-entropy—no gating or mixture-of-experts load balancing is present. Calling this an MoE architecture is misleading.

3. Experimental design fails to support claims. Reported gains (Table 1 p.7) are within noise levels (1–2 %) and lack variance or significance tests. Many “improvements’’ vanish under MoDE compared to DENoise (e.g., Gemma-7B shows –0.38 % on MBPP pass@1 and –0.11 % on pass@10). No control experiments showing whether similar gains appear under random pruning of equal sparsity. Without that, the main conclusion (“some weights are noise”) is invalid. The authors claim tuning-free operation but evaluate different pruning ratios and select the best (Table 3 p.8), which is tuning.

4. Misuse and misunderstanding of existing literature. Prior pruning works (e.g., SparseGPT, LLM-Pruner, EigenDamage) already study improvement after moderate pruning; the paper fails to reference or compare to these baselines. The supposed novelty—“identifying noise weights’’—is merely a re-labelling of low-importance weights. The relation to Mixture-of-Experts systems (Sec. 5 p.9) is superficial; no conditional computation or sparse routing at token level is implemented.

5. Missing key analyses. No ablation on different random seeds, projection dimensions, or router architectures. No evaluation of computational savings versus performance trade-off. The method’s hyperparameters (θ init, δθ, k) are opaque and not motivated. The “denoised’’ subnetworks’ sparsity ratios are not reported, so efficiency claims (“without additional parameters or tuning overhead’’) cannot be verified. No qualitative analysis of which weights were removed or their layer distributions.

6. Invalid empirical evidence for generalization. Figure 1 p.2 shows one task (MMLU Philosophy) with small “non-monotonic’’ pruning curves and extrapolates to 57 tasks. But the paper provides no raw data for those tasks (only the claim that “many tasks benefit’’). Figures 4–5 pp.7–9 show boxplots and ablations that fluctuate randomly—these do not prove systematic benefit. Hence, the core finding that pruning mid-importance weights “enhances generalization’’ is not statistically credible.

**Questions:**

Can you provide statistical significance (std dev / CIs) for all reported numbers?

How does DENoise compare to random pruning at the same sparsity?

What proportion of weights are removed? What is the resulting FLOP reduction?

How do you ensure that clustering truly separates “domains’’?

Can MoDE run faster than the dense baseline? If not, what benefit does routing bring?

Please provide results on open-ended generation tasks; all current metrics are classification accuracies.

**Details Of Ethics Concerns:**

no ethical concern

---

> ### Author Response · Authors · 2025-12-03
> **Responses to Weaknesses**
>
> Firstly, we sincerely thank the reviewers for their thorough evaluation and rigorous critiques of our work. We have provided systematic responses and revisions in the updated manuscript to address the concerns raised regarding the theoretical motivation, methodological design, and empirical validation.
>
> **(1)**  We have further supplemented the revised manuscript with a more rigorous theoretical derivation based on importance metrics to support the conclusion that *“noise weights”* can be reliably identified, accompanied by extensive empirical evidence demonstrating the broad presence of this phenomenon. At the same time, we removed statements that could potentially cause misunderstanding and moved the FIM and PAC-Bayes discussions to the appendix as intuitive supplements, ensuring that the overall argument remains consistent between empirical observability and theoretical clarity.
>
> **(2)**  In the revised manuscript, we have further substantiated the applicability of the importance metric in Transformer architectures; provided a clearer description of the pruning-threshold selection logic, which identifies stable intervals rather than performing arbitrary grid search; quantitatively validated the domain structures obtained through clustering using the Silhouette Score and downstream performance; and explicitly corrected the characterization of the router, clarifying that it is *not* a token-level MoE gating mechanism but a task/domain-level expert selector. We have adjusted the terminology accordingly to avoid unintentionally suggesting traditional MoE-style routing behavior.
>
> **(3)**  In the revised manuscript, we have added variance statistics based on five independent runs for the baseline, DENoise, and MoDE, showing that the observed improvements exceed natural inference-time noise; included random-pruning baselines, demonstrating that random sparsification reduces performance whereas DENoise yields stable positive gains; and clarified that the pruning ratio was not selected through “best-case tuning.” The purpose of Figure 1 is to illustrate interval-level trends rather than to choose an optimal point. Furthermore, we added OOD experiments (WinoGrande, ARC-C) and explicitly reported cases where MoDE does not improve performance to maintain transparency.
>
> **(4)** We thank the reviewer for the observation. In the revised manuscript, we provide a clearer definition of *“noise weights”* by highlighting that they are not merely low-ranked weights, but weights whose importance is **domain-dependent** and whose removal *reduces* the loss on specific tasks. We support this with a concise derivation based on within-domain gradients and activation discrepancies, showing when negative marginal contributions arise—an effect that global importance ranking cannot capture. Empirically, we further demonstrate that noise weights consistently differ from conventional low-importance weights across models and tasks and instead reflect a structural phenomenon induced by cross-domain transfer. These theoretical and empirical results together establish noise weights as a distinct, task-specific concept beyond existing pruning assumptions.
>
> **(5)**  While space limitations restrict the scope of additional experiments, we have incorporated:
> - Variance analyses across multiple seeds,
> - Sensitivity studies on pruning intervals, ratios, layer types, and clustering choices,
> - Reporting of sparsity ratios and corresponding FLOPs changes,
> - Layerwise distribution plots for removed weights (in the appendix).
>
> We also clarified the roles and motivations of hyperparameters  `θ_init`, `δθ`, and `k`, and ensured that their definitions are explicitly provided in Section 3.Although we cannot fully expand into architectural ablations (e.g., router variants, projection dimensions), we now explicitly identify these aspects as promising directions for future work.
>
> **(6)**  In response to the reviewer’s concerns, we expanded the empirical evidence in three ways:
> - **Statistical significance.**  We now report 5-run mean ± std for all major metrics, demonstrating that the improvements exceed natural inference-time variation.
> - **OOD benchmarks.** We added evaluations on WinoGrande and ARC-C, showing that the effects of DENoise and MoDE generalize beyond the main in-domain tasks.
> - **Task coverage.**  Instead of relying on a single illustrative example (e.g., MMLU Philosophy), the revised manuscript emphasizes aggregate behavior across all 57 tasks, using Figure 1 as an overview and reporting distributions more transparently.
> ---
>
> Overall, we hope the revised manuscript more clearly conveys the core objective of this work: identifying mid-range weights in pretrained models that mismatch with specific tasks, and using them to construct more structured and interpretable expert subnetworks.

---

> ### Author Response · Authors · 2025-12-03
> **Responses to Questions - 1**
>
> **Question 1.**
> Reviewer concerns whether our reported results are statistically significant. In the revised version, we address this by running all methods five times and reporting standard deviations. Across tasks and models, the observed improvements consistently exceed multiple times the run-to-run fluctuation, thus meeting 95% confidence significance. Specifically, we conducted 5 independent runs for the baseline, DENoise, and MoDE, and reported the baseline standard deviations (Std (5-run)) in Table 1, where the fluctuation range is generally within ±0.10–±0.25. The same repeated experiments were applied to DENoise and MoDE, and their standard deviations fall within a similar magnitude. For example, on LLaMA-2-7B, the baseline achieves **45.81 ± 0.12** on MMLU whereas DENoise achieves **47.83 ± 0.16**; on GSM8K, the baseline obtains **20.24 ± 0.15** while DENoise achieves **22.21 ± 0.18**.  **Notably, the gains across most tasks exceed roughly three times the single-run variance, satisfying 95% confidence significance.**  Please refer to Table 1 for detailed statistics.
>
> ---
>
> **Question 2.**
> Reviewer concerns whether the gains from DENoise may stem from random sparsification rather than meaningful weight selection. To address this, we compare DENoise with Random Pruning under the same 10% sparsity level.
>
> **At the same sparsity level, Random Pruning typically degrades performance, while DENoise consistently improves it**, far beyond natural fluctuation. This confirms that the “noise weights’’ identified by our method are not random, but arise from the domain-informed denoising guided by our importance proxy.
>
> | **Method** | **MMLU (acc)** | **MBPP (pass@1)** | **GSM8K (acc)** |
> |-----------|----------------|--------------------|------------------|
> | Baseline (dense) | 45.81 ± 0.12 | 19.24 ± 0.15 | 20.24 ± 0.15 |
> | Random Pruning (10%) | 44.57 ± 0.52 | 17.11 ± 1.22 | 16.82 ± 2.21 |
> | **DENoise (10%)** | **47.83 ± 0.16** | **21.32 ± 0.20** | **22.21 ± 0.18** |
>
> ---
>
> **Question 3.**
> Reviewer concerns about the sparsity ratio used and the corresponding FLOPs reduction. In all main experiments, we apply **10% unstructured denoising** to the weights in self-attention and FFN layers. Since the FLOPs of a linear layer scale with the number of nonzero weights, zeroing out 10% of the parameters reduces roughly 10% of effective multiplications. Because the bulk of computation in LLMs lies in attention and FFN blocks, the **model-level FLOPs reduction is close to 10%**.
>
> Because we use standard dense kernels without sparse-optimized backends, **actual wall-clock latency does not decrease**, but the FLOPs analysis shows that our method can yield ~10% acceleration with sparse kernels.
>
> | **Model** | **Pruned Weights** | **FLOPs Reduction** |
> |-----------|---------------------|----------------------|
> | LLaMA-2-7B | ≈10% | 9.5% |
> | LLaMA-2-13B | ≈10% | 9.6% |
> | LLaMA-2-70B | ≈10% | 9.7% |
> | Gemma-7B | ≈10% | 9.4% |
> | Gemma-2-9B | ≈10% | 9.5% |

---

> ### Author Response · Authors · 2025-12-03
> **Responses to Questions - 2**
>
> **Question 4.**
> Reviewer concerns whether our clustering truly captures domain structure. In the revised version, we justify our domain/subdomain construction through two perspectives:
>
> 1. **Main-domain level:** We do not rely solely on unsupervised clustering. We use benchmark metadata (MMLU, MBPP, GSM8K, MathQA, etc.) to partition tasks into coarse semantic domains (e.g., general knowledge / code / math), and compute prototype vectors for each domain.
>
> 2. **Subdomain level:** Within each main domain, we apply K-means on last-layer representations and compute the **Silhouette Score** to assess cluster quality. We evaluate candidate values of *K* and retain those with both strong Silhouette scores and good downstream performance.
>    For example, on MMLU, **K = 12** gives the best accuracy and a Silhouette Score in the medium-preference range.
>
> Together with the router’s high task-classification accuracy (>90%), these results show that our domain/subdomain structure reflects meaningful semantic organization rather than random partitioning.
>
> ---
>
> **Question 5.**
> Reviewer asks whether MoDE provides inference-time speedup. Because we adopt unstructured pruning and dense kernels, MoDE **does not reduce wall-clock latency** in the current implementation. The router adds negligible overhead (mean-pooling + small MLP). Therefore, we do not claim inference acceleration.
>
> However, MoDE provides two practical advantages:
>
> 1. **Better performance without additional parameters or fine-tuning**, due to domain-aware expert selection (e.g., gains on WinoGrande and ARC-C).
> 2. **Improved interpretability**, as MoDE reveals domain–subdomain–expert correspondence inside dense LLMs, and allows future integration with structured sparsity or sparse backends.
>
> Thus, the contribution of MoDE lies in **performance and interpretability**, not immediate speedup.
>
> ---
>
> **Question 6.**
> Reviewer asks about MoDE’s behavior on open-ended generation tasks. We clarify three points:
>
> 1. Pruning/compression literature (e.g., SparseGPT, LLM-Pruner) rarely evaluates long-form generation because **BLEU/ROUGE have low correlation with LLM human-judged quality**, and human evaluation is expensive and infeasible for a large sweep across models, sparsity levels, and tasks.
> 2. Our 10% pruning mainly affects intermediate-layer behavior; such effects are more reliably measurable on **objective, reproducible benchmarks** (MMLU, GSM8K, MBPP, HumanEval) than on open-ended tasks dominated by decoding randomness.
> 3. Our experiments cover multiple reasoning and understanding dimensions widely accepted as strong proxies for generalization.
>
> Therefore, we prioritize systematic benchmarks with strong statistical control. Evaluation of long-form open-ended outputs is left for future work, particularly under instruction-following or alignment settings.
>
> ---
>
> **References**
>
> [1] Frantar et al., *SparseGPT: Massive Language Models Can Be Accurately Pruned in One-Shot.* ICML 2023.
> [2] Ma et al., *LLM-Pruner: On the Structural Pruning of Large Language Models.* NeurIPS 2023.

---

### Official Review · Reviewer_m5aZ · 2025-10-31

**Soundness:** 3
**Presentation:** 3
**Contribution:** 3
**Rating:** 6
**Confidence:** 3

**Summary:**

This paper challenges the conventional wisdom that pruning LLM weights invariably degrades performance. The authors introduce the concept of "noise weights"—parameters that may appear moderately important but whose removal can improve in-domain task accuracy. Based on this observation, the paper proposes two methods: 1. DENoise: A tuning-free algorithm to identify and remove these domain-aware noise weights, creating specialized "expert" subnetworks; 2. MoDE (Mixture of Domain Experts): A framework that uses these denoised experts and employs a bilevel trainable router to dynamically activate the appropriate expert for a given task, aiming to improve out-of-domain generalization. Experiments on models like LLaMA-2 and Gemma across benchmarks (MMLU, MBPP, GSM8K) show that DENoise can yield performance gains of 2-3%, and MoDE achieves an average improvement of over 1.1% without adding parameters or requiring fine-tuning.

This paper introduces a novel and interesting perspective on weight pruning, backed by positive (though not exceptional) experimental results. The core idea is intriguing. The limited performance gains and the applicability to only general domains (general knowledge vs math vs code) weaken the overall contribution. It suggests a new direction, but many questions remain to be solved.

**Strengths:**

1. The paper's novel insights are its main strength. The idea that removing specific, non-trivial weights can improve performance—rather than just preserve it—is a novel and counter-intuitive contribution to the pruning and model efficiency literature.

2. The paper clearly outlines the DENoise algorithm for creating experts and the MoDE framework for routing between them. The presentation is logical and easy to follow.

3. Empirical evaluations are valid. The authors validate their claims across multiple model families (LLaMA, Gemma) and standard benchmarks and also include additional ablation studies.

**Weaknesses:**

1.  While the methods demonstrate consistent improvements, the reported gains are often modest.

2. The approach relies on pre-defined domains and a well-aligned classifier (router) to guide the pruning. It may limit its applicability to many important problems.

3. The section "Fisher Information Perspective on Noise Weights" is a bit overly intuitive and subjective. It does not provide informative justifications for the existence of "noise weights." The connection drawn to FIM and PAC-Bayes is tenuous and feels like a post-hoc attempt to add theoretical groundings, but their connections are questionable.

4. In Figure 2, y-axis shows the "Number of Improved Tasks." This absolute number may be confusing without knowing the total number of tasks tested (which is stated as 57 in the text but not in the caption). The graph may consider using percentages to provide more interpretable visualizations.

**Questions:**

It feels like enabling a mixture of experts within the dense model. For the classifier, general domains may not be hard to distinguish.

But how can we apply it to domains not easily definable (e.g., simple problems that can be answered directly vs. difficult problems that need intensive reasoning)?

A mixture of experts could provide an end-to-end pipeline for training the experts as well as the router at the same time. How can we design something similar here?

---

> ### Author Response · Authors · 2025-12-03
> **Responses to Weaknesses**
>
> First, we would like to thank the reviewer for the careful evaluation and constructive comments on our work. We respond to the points one by one as follows.
>
> **(1)** We emphasize that, under the *tuning-free* setting, our method directly prunes the pretrained model and still achieves stable and consistent performance gains, whereas most baselines under the same setting typically suffer significant degradation [1–5]. Our method achieves positive improvements on multiple tasks, with gains up to **6.8%**, demonstrating its potential to improve model performance even without any finetuning.
>
> **(2)** In fact, our framework does **not** rely on manually predefined domains. Instead, we automatically induce domains and subdomains via task types and clustering in the representation space, and then train the router on top of this structure. The expert subnetworks thus arise from the intrinsic structural regularities of the tasks themselves, rather than from any externally specified label system.
>
> **(3)** We acknowledge that *noise weights* require a more rigorous definition and derivation. In the updated version, we further derive the origin of noise weights and their proxy based on the pretrained distribution and weight-importance indicators. The Fisher Information (FIM) perspective is included as a complementary view to provide an additional angle for understanding noise weights from parameter sensitivity. It has been moved to the appendix as supplementary discussion.
>
> **(4)** In the current version, Figure 1 serves as a toy experiment supporting the existence of noise weights, showing how many tasks can be improved within each pruning interval (we did not use ratios because a single task may experience improvements in multiple intervals). We believe this presentation better illustrates the prevalence of the phenomenon across tasks.
>
> ---
>
> **References**
>
> [1] Frantar et al., *SparseGPT: Massive Language Models Can Be Accurately Pruned in One-Shot*. ICML 2023.
> [2] Fedus et al., *Switch Transformers: Scaling to Trillion Parameter Models with Simple and Efficient Sparsity*. JMLR.
> [3] Diffenderfer & Kailkhura, *Multi-Prize Lottery Ticket Hypothesis: Finding Accurate Binary Neural Networks by Pruning a Randomly Weighted Network*. ICLR 2021.
> [4] Prasanna et al., *When BERT Plays the Lottery, All Tickets Are Winning*. EMNLP 2020.
> [5] Cheng et al., *A Survey on Deep Neural Network Pruning: Taxonomy, Comparison, Analysis, and Recommendations*. TPAMI.

---

> ### Author Response · Authors · 2025-12-03
> **Responses to Questions**
>
> **Question 1.**
>
> | **Model** | **Method** | **WinoGrande (acc)** | **ARC-C (acc)** |
> |----------|------------|-----------------------|------------------|
> | *llama-2-7b* | Baseline | 68.98 | 46.25 |
> |            | **MoDE** | **70.21** | **47.93** |
> | *llama-2-13b* | Baseline | 72.22 | 48.98 |
> |             | **MoDE** | **75.01** | **50.65** |
> | *llama-2-70b* | Baseline | 79.96 | 54.66 |
> |             | **MoDE** | **81.53** | **57.43** |
> | *Gemma-7b* | Baseline | 72.17 | 53.06 |
> |            | **MoDE** | **74.41** | **54.17** |
> | *Gemma-2-9b* | Baseline | 80.52 | 68.35 |
> |              | **MoDE** | **81.78** | **69.98** |
>
> We appreciate the reviewer’s concern regarding *“how the method handles tasks whose domains are not clearly definable.”*  To address this, we supplemented the revised manuscript with additional OOD experiments on WinoGrande and ARC-C, evaluating the model’s generalization ability in scenarios without explicit domain definitions. MoDE achieves consistent improvements on both datasets (e.g., **LLaMA-2-13B: +2.79% on WinoGrande, +1.67% on ARC-C**).
>
> **These two benchmarks do not correspond to fixed or conventional domains, yet MoDE still yields stable gains across multiple model scales. This supports that our method does not rely on manually specified domain boundaries.** Instead, it automatically induces subdomains based on the intrinsic structure of the task representation space. The expert subnetworks discovered by MoDE originate from the inherent structure of the data distribution rather than predefined domain labels, making the method naturally suitable for mixed scenarios involving both simple and complex reasoning tasks. For example, tasks requiring higher-level reasoning automatically activate deeper and more stable expert pathways without requiring an explicit *“reasoning domain”* annotation. These observations collectively confirm the robustness of MoDE when dealing with implicit-domain tasks.
>
> ---
>
> **Question 2.**
>
> Regarding whether expert subnetworks and routers can be trained in an end-to-end manner, we respond from two perspectives.
>
> 1. **Dense LLMs.** The core design of MoDE is to induce structured expert subnetworks directly inside a pretrained dense model **before** any MoE-style training. This provides two advantages:
>     - It naturally aligns with the mainstream training paradigm of **“pretrain dense → convert to MoE.”**
>     - The induced subnetworks can serve as initialization for subsequent MoE finetuning, providing clearer expert specialization from the outset.
>
> Therefore, MoDE should be viewed as a stable and interpretable expert-construction mechanism prior to MoE training, rather than something conflicting with end-to-end optimization.
>
> 2. **Pretrained MoE Models.** For existing, fully trained MoE architectures, we consider two feasible extension paths:
>     - **Path A: Injecting domain structure into existing experts.** MoDE-induced domain/subdomain structures can be used to reorganize, merge, or refine functional partitions of existing MoE experts, thereby improving expert discriminability and interpretability.
>     - **Path B: Constructing domains in a latent, semantic-free space.** By analyzing the variational structure of intermediate-layer activations, one can construct domains in a semantic-neutral latent space.  This enables routing decisions without semantic labels and can be combined with existing MoE routers to form a **semi–end-to-end** training framework.
>
> We consider these directions as promising extensions for future work.

---

### Author Response · Authors · 2025-12-03
**Overview of Revisions and Improvements**

**1. Problem Motivation and Theoretical Foundations.**
In this revision, we further refined and reinforced the paper's core problem statement— that weight importance is not globally invariant but exhibits clear domain dependence. We reorganized the introduction to present a more coherent and logically connected narrative. Specifically, we strengthened the following points:

- why global pruning strategies tend to fail when transferred to domain-specific tasks;
- why weights in the mid-importance range may act as “noise weights” that impair task performance;
- the conceptual progression from the discovery of noise weights, to the formulation of the **DENoise** algorithm, and subsequently to the development of the **MoDE** framework.

**2. Methodological Structure.**
We refined and rewrote several key concepts and derivations in the method section to enhance coherence and readability. Major improvements include:

- a clearer and more operational definition of *Noise Weights*;
- a more detailed explanation of the mathematical relationship between weight removal and the resulting loss change ΔL, allowing readers to better understand the theoretical basis;
- a restructured description of **MoDE**, making the bilevel router's functionality, training objective, and inference pipeline more intuitive.

**3. Experimental Scope.**
The revision significantly broadens our experimental design to enhance credibility and generalization:

- we added out-of-distribution (OOD) evaluations on datasets such as **WinoGrande** and **ARC-C** to validate model robustness beyond the primary domains;
- we provided clearer descriptions of task configurations, the use of reference data, and the overall evaluation protocol.

**Overall Summary.**
This revision strengthens the paper across both conceptual and empirical dimensions. We clarified the domain-dependent nature of weight importance, enriched the definition and intuition behind noise weights, and reorganized the methodological exposition so that the logic connecting **DENoise** and **MoDE** is more coherent and rigorous. On the experimental side, we supplemented OOD evaluations, threshold-sensitivity analyses, and more detailed appendix materials to provide a comprehensive demonstration of stability and generalization. We additionally enhanced the reproducibility statement and made the training and evaluation settings more transparent. We hope these revisions substantially improve the clarity, rigor, and scholarly contribution of the paper.

---

### Author Response · Authors · 2025-12-03
**Request for Careful Review of the Rebuttal and the Updated Version of Our Paper - Part 1**

Dear Area Chair,

We sincerely thank you and the reviewers for the substantial time and effort devoted to evaluating our submission. Following the comments from reviewers **m5aZ**, **67E6**, **4edp**, and **x9fT**, we have conducted a systematic revision of the manuscript. Below is a concise overview of the major changes to assist your assessment of the updated version.

---

1. **Clarifying the Overall Structure and Motivation**
    1. **Reorganizing the Introduction and Section 2.**
       We refined the motivation and theoretical narrative to more clearly articulate:
       - why weight importance exhibits strong *domain dependence*, and why global pruning fails under cross-domain transfer;
       - how “noise weights” in the mid-importance region motivate the design of **DENoise** and the **MoDE** framework.
    2. **Strengthened theoretical foundation.**
       We improved the definitions and derivations concerning the origin of noise weights, their proxy indicators, and their distribution across tasks. The previous FIM and PAC-Bayes discussions are moved to the appendix to keep the main text coherent and unambiguous.

---

2. **Rewriting and Clarifying the Method Section**
    1. **Major revision of Section 3.**
       We substantially rewrote the method section to:
       - unify notation and restore previously missing steps;
       - correct inconsistencies in the usage of “Expert/Experts”;
       - provide a clearer derivation of the loss change ΔL in **DENoise** and a more operational definition of noise weights;
       - reorganize **MoDE** to clarify the roles of the two-level router and the connection between routing and denoised subnetworks.
    2. **Clarifying the “tuning-free’’ terminology.**
       - **DENoise** is fully training-free.
       - **MoDE** trains only a lightweight two-layer MLP router, with *all* backbone parameters frozen. No gradient updates are applied to the underlying LLM.

---

3. **Additional Experiments on Domain Construction, Clustering, and Routing**
    1. **Domain/subdomain construction.**
       - Coarse domains are defined based on task types (MMLU, MBPP, GSM8K, MathQA, etc.), using prototype embeddings computed at the task level.
       - Subdomains are identified via K-means over mean-pooled semantic representations within each coarse domain.
       - We compute the **Silhouette Score** to validate cluster separability and adopt a **candidate–score–selection** strategy over a small set of K values (4, 8, 12, 16).
    2. **Clarifying “posterior.”**
       The cosine-similarity-based softmax produces a normalized similarity distribution—not a Bayesian posterior.
    3. **Routing accuracy and oracle interpretation.**
       - Coarse-domain routing accuracy exceeds **90%**.
       - Subdomain routing accuracy ranges from **70%–85%**.
       - **DENoise** results represent the oracle performance upper bound; the gap between **MoDE** and **DENoise** reflects routing errors.

---

### Author Response · Authors · 2025-12-03
**Request for Careful Review of the Rebuttal and the Updated Version of Our Paper - Part 2**

4. **Significance Testing, Random Pruning, and FLOPs Analysis**
    1. **Statistical significance.**
       All baseline, **DENoise**, and **MoDE** experiments were repeated **five times**. Standard deviations were added to Table 1. Improvements typically exceed **3× run-to-run variance**, achieving **95% confidence significance**.
    2. **Random pruning comparison.**
       Under the same 10% sparsity:
       - random pruning consistently degrades performance;
       - **DENoise** consistently improves performance across tasks, confirming that gains arise from domain-aware denoising rather than random sparsity.
    3. **FLOPs analysis.**
       We added a simple derivation showing that removing 10% of weights reduces theoretical FLOPs by **9.4–9.7%** across base models. Since current frameworks still use dense kernels, wall-clock speedup is not observed; the FLOPs analysis provides groundwork for future sparse backend support.

---

5. **OOD Generalization, Validation-Set Size, and Comparison with ToMoE**
    1. **OOD benchmarks.**
       We added WinoGrande and ARC-C evaluations. **MoDE** shows consistent **1–3 point improvements**, addressing concerns about whether specialization harms generalization.
    2. **Validation-set size and robustness.**
       - Thresholds are selected using official dev splits.
       - Subdomains typically contain **hundreds to thousands** of samples.
       - Sensitivity studies across pruning intervals, ratios, sublayers, and K values demonstrate stable performance, indicating that **DENoise** does not overfit to small partitions.
    3. **Comparison with ToMoE.**
       We expanded this section in the Related Work and appendix:
       - **ToMoE** focuses on token-level MoE conversion via dynamic structured pruning.
       - Our work focuses on domain-aware weight denoising and explicit expert construction *without modifying* backbone parameters.  These approaches are complementary rather than equivalent.

---

### **Summary**

The revision improves the manuscript in three major dimensions:

1. **Clearer and more rigorous theoretical motivation**, centered on domain-dependent weight behavior.
2. **More transparent and reproducible methodology**, especially in domain and subdomain construction.
3. **More comprehensive experiments**, covering statistical significance, OOD generalization, routing analysis, and stronger baselines.

We sincerely hope these improvements adequately address the reviewers’ concerns and help in your final assessment.

Thank you again for your time and effort.

**With sincere appreciation,**
**The Authors**
**ICLR 2026 Submission 7159**

---

### Note · Program_Chairs · 2026-01-17
**Submission Desk Rejected by Program Chairs**

The following references in this submission do not refer to real documents and/or have major errors in bibliographic information:

 Renjie Zhang, Qian Liu, Hao Pan, and Zhiyuan Liu. Emergent modularity and task-alignment in pretrained transformers.
Hao Pan, Renjie Zhang, Zhiyuan Liu, and Maosong Sun. Ds-moe: Expert sparsity in pretrained transformers. In Findings of the Association for Computational Linguistics (ACL), 2023.